# SMASH v1.0: A Differentiable and Regionalizable High-Resolution Hydrological Modeling and Data Assimilation Framework

François Colleoni[1], Ngo Nghi Truyen Huynh[1], Pierre-André Garambois[1], Maxime Jay-Allemand[2], Didier Organde[2], Benjamin Renard[1], Thomas De Fournas[1], Apolline El Baz[1], Julie Demargne[2], and Pierre Javelle[1]

[1]INRAE, Aix-Marseille Université, RECOVER, 3275 Route Cézanne, 13182 Aix-en-Provence, France
[2]HYDRIS Hydrologie, Parc Scientifique Agropolis II, 2196 Boulevard de la Lironde, 34980 Montferrier-sur-Lez, France

**Correspondence:** Pierre-André Garambois (pierre-andre.garambois@inrae.fr)

**Abstract.** The `smash` software is a differentiable and regionalizable framework enabling modular high-resolution hydrological modeling and data assimilation, from catchment to regional and country scales, for water research and operational applications. `smash` combines various process-based conceptual operators for vertical and lateral flows, which can be hybridized with a descriptors-to-parameters neural network for regionalization. `smash` features an efficient, differentiable Fortran solver using `Tapenade` to automatically derive the adjoint model that supports CPU forward-inverse parallel computing and spatially distributed optimization of large parameter vectors thanks to accurate cost gradient, interfaced in Python using `f90wrap`. This article presents `smash` algorithms, their open-source code, documentation and tutorials. It highlights foundational research, benchmarking on state-of-the-art datasets, and readiness for scientific and operational use. To ensure reproducibility, open-source datasets are used to demonstrate the main functionalities of `smash`, including parallel computation performances and the application of multiple spatially distributed conceptual model structures over a large catchment sample. These functionalities include uniform or spatially distributed calibration and regionalization by learning the relation between descriptors and parameters. Provided Python tool allows application to any other catchment from globally available datasets. Using CAMELS, as per recent articles, median $KGE > 0.8$ are obtained in local spatially distributed calibration for daily GR-like and VIC-like model structures at $dx = 1'30''$ ($\sim 3km$) and $KGE > 0.6$ in spatio-temporal validation in a regionalization context. The regionalization of a high resolution hourly GR-like model structure at $dx = 500m$ over a difficult mediterranean flash-flood prone case results in $NSE > 0.6$ in spatio-temporal validation. The proposed differentiable and regionalizable spatially distributed modeling framework is designed for gradient-based variational data assimilation, applicable to initial states (not shown) and parameters estimation at multiple time scales, and is intended for collaborative research and operational applications. Additionally, `smash` supports the implementation of other differentiable hydrological and hydraulic models, as well as hybrid physics-AI models, further enhancing its versatility and applicability.

## 1 Introduction

Hydrological models are indispensable tools for hydrosystems functionning understanding, floods and low flows forecasting, sustainable water management and infrastructure design, environmental protection, and adaptation to a changing climate.

Indeed, measurements of hydrological responses are not ubiquitously available (e.g. Beven (2011)) while "everywhere relevant" (Bierkens et al., 2015) estimation of hydrological state-fluxes is expected. A model is hence needed to extend and predict those quantities of interest based on available data.

High-resolution spatial datasets have become increasingly accessible, often on a global scale, and enable describing topography-soil-vegetation properties as well as atmospheric variables. Examples include the ECMWF atmospheric reanalysis version 5 (ERA5) (Hersbach et al., 2020) and rainfall product MSWEP (Beck et al., 2019), flow directions IHU (Eilander et al., 2021) from MERIT terrain elevations (Yamazaki et al., 2017), the SoilGrids pedology (Hengl et al., 2017), daily discharge from Caravan-CAMELS (Kratzert et al., 2023; Addor et al., 2017), that will be used hereafter. Such data can be directly exploited by grid based spatially distributed hydrological models, whose development at "hyper-resolution" ($1km^2$ or finer) is recognized as a "grand challenge for hydrology" to address water problems facing society (Wood et al., 2011; Bierkens et al., 2015).

Hydrological responses result from combined non-linear vertical and lateral physical processes occurring at multiple scales in the critical zone and their limited observability (e.g. Beven, 1989; Milly, 1994; Blöschl and Sivapalan, 1995; Refsgaard, 1997; Vereecken et al., 2019) makes hydrological modeling uncertain and difficult (e.g. Liu and Gupta, 2007)). In the absence of directly exploitable first principles in hydrology (e.g., Dooge (1986)), as opposed to flow mechanistic equations in continuous media such as river hydraulics, meteorology or oceanography, and given the high heterogeneities of continental hydrosystems compartments and the lack of "scale-relevant theories" (Beven, 1987), process-based hydrological models generally include a certain amount of empiricism. It represents an avenue for the fusion of data assimilation (DA) and uncertainty quantification (UQ) with machine learning (ML) and deep learning (DL) techniques to better exploit the informative richness of multi-source data.

The differentiability of the forward numerical model is a key enabler for gradient-based optimization of high-dimensional parameter vectors. For example, in variational data assimilation for 1D or 2D hydraulic models (Monnier et al., 2016; Brisset et al., 2018), or in spatialized hydrology (Castaings et al., 2009; Jay-Allemand et al., 2020). While differentiability may appear unnecessary for simple lumped hydrological models with only a few parameters, where sampling-based calibration or gradient-free methods remain efficient, the situation changes drastically for spatially distributed models involving thousands of parameters. In such high-dimensional settings, exhaustive sampling becomes computationally infeasible. Numerical differentiability enables the computation of accurate gradients of the cost function or model outputs with respect to high-dimensional parameters, thereby facilitating the use of efficient gradient-based optimization methods. This is particularly important when coupling physical models with neural networks requiring accurate gradients, as demonstrated in recent work on learnable regionalization (Huynh et al., 2024b) and internal flux correction (Huynh et al., 2024a), with large-scale evaluations in (Huynh et al., 2025). These approaches rely on numerically differentiable solvers and accurate gradients enabling to train thousands of parameters effectively. This perspective aligns with Shen et al. (2023), who emphasizes the importance and potential of differentiable modeling in geosciences, highlighting how it can enhance learning, inference, and integration of physical knowledge within hybrid modeling frameworks.

The "resolution-complexity continuum" (Clark et al., 2017) has been explored over the past five decades through various modeling approaches, ranging from point-scale processes numerically integrated at larger scales to spatially lumped repre-

sentations of system responses (Hrachowitz and Clark, 2017). Among the diverse hydrological models and their underlying
hypotheses, components generally describe water storage and transfer (e.g. Fenicia et al. (2011)) through various combinations
and parameterizations of vertical and lateral storage-flux operators. Several model comparison experiments have analyzed differences between various modeling approaches, evaluating performance in terms of streamflow modeling (Perrin et al., 2001; Reed et al., 2004; Duan et al., 2006; Orth et al., 2015) and internal states such as soil moisture (Orth et al., 2015; Bouaziz et al., 2021). Orth et al. (2015) concluded that "added complexity does not necessarily lead to improved performance of hydrologi-
cal models". Notably, parsimonious conceptual models, whether lumped or semi-lumped, have performed efficiently in large sample studies (e.g. GR model in Perrin et al. (2001), GRSD model in De Lavenne et al. (2019), GR and MORDOR models in Mathevet et al. (2020), FUSE models in Lane et al. (2019) and references therein). Large sample studies have also been undertaken with spatially distributed models among which VIC (Mizukami et al., 2017) with a Multiscale Parameter Regionalization (MPR) (Samaniego et al., 2010) or with pixel-wise calibration on global maps of streamflow characteristics (Yang et al., 2019),
a gridded version of HBV applied with MPR like descriptors-to-parameters regressions on a global dataset (Beck et al., 2020), GloFas (Hirpa et al., 2018), NHM (Towler et al., 2023), Wflow (Aerts et al., 2022; van Verseveld et al., 2024) or runoff relevant parameters of E3SM using a surrogate-assisted Bayesian framework (Xu et al., 2022). Differentiable numerical hydrological modeling has made significant progress in recent years (spatially distributed VDA in Castaings et al. (2009); Lee et al. (2012); Jay-Allemand et al. (2020)), for large catchment sample studies with hybrid physics-AI, both with lumped approaches (e.g.,
Feng et al. (2024)) and with high-resolution, spatially distributed frameworks (Huynh et al., 2024b, 2025). These large sample studies enable more general and statistically sound analyses of model performances (Andréassian et al., 2009; Gupta et al., 2014), addressing large-scale challenges with consistent methodologies across various scales and conditions.

All hydrological models are inherently conceptual and calibration or learning is generally required due to limitations and uncertainties in their structure, parameter representativity, data availability and initial and boundary conditions. These models
are typically calibrated and validated using discharge time series at the catchment outlet(s) (Sebben et al., 2013). However, calibrating hydrological model parameters from sparse and integrative discharge data is a challenging inverse problem complicated by equifinality issues (Bertalanffy, 1968; Beven, 1993, 2001) especially for distributed models with a large number of cells and parameters ("curse of dimensionality"). Using spatially uniform parameters may not be the best way to exploit a spatially distributed model (under parameterization), while fully distributed parameters calibration, which requires a gradient
based approach (Castaings et al., 2009; Lee et al., 2012; Jay-Allemand et al., 2020), is facing over parameterization. Therefore, a parameter regionalization approach using multilinear descriptors-to-parameters transfer functions has been proposed for distributed models (Beck et al., 2020). More recently, this approach has been advanced with regionalization neural networks (Huynh et al., 2024b) integrated into the differentiable spatialized `smash` model (Colleoni et al., 2022), which is the focus of the present article introducing a new numerical code and conducting original tests on a large sample of catchments with open
source data. This approach also enables learning via cost functions based on hydrological signatures, which are obtained using automatic signal analysis algorithms applicable to large samples with `smash` (Huynh et al., 2023).

This article presents the computational framework `smash` dedicated to *Spatially distributed Modeling and ASsimilation for Hydrology*. The `smash` framework combines vertical and lateral flow operators, either process-based conceptual or hybrid with

neural networks (allows learning regionalization relations between descriptors and parameters), and perform high dimensional
optimization from multi-source data. It is based on an efficient and automatically differentiable Fortran solver enabling CPU parallel computing, that is interfaced in Python using `f90wrap` (Kermode, 2020; Jay-Allemand et al., 2022). This open-source `smash` code, in its version v1.0 (https://github.com/DassHydro/smash), is presented here in terms of mathematical formulation, numerical modeling approach and functionalities while full details can be found in our research articles from which this software stems (Colleoni et al., 2022; Huynh et al., 2023, 2024b) and in the online documentation (https://smash.recover.inrae.fr). Note that `smash` has also been developed for operational applications. It is the core solver of the French flash flood forecasting system (Piotte et al., 2020). The proposed framework leverages adjoint-based VDA, enabling the simultaneous inference of high-dimensional and spatially distributed parameters (as illustrated) and initial states (implementation available and tested in `smash` v1.0 but not shown), applicable at both long and short time scales.

This article is organized as follows. Section 2 describes the `smash` forward model and the inverse algorithm. In section 3 we describe the `smash` build system framework, documentation and computational performance. Some applications of `smash` are demonstrated in section 4 using open-source datasets focusing on the contiguous United States (CONUS) and on a high-resolution flash flood-prone case study in France. Section 5 illustrates other aspects of `smash` not presented in section 4, followed by conclusions in section 6.

## 2   Model and optimization algorithms description

The `smash` framework contains various hydrological model structures with varying vertical and lateral flow operators as well as spatialized routing schemes. It is designed to simulate discharge hydrographs and hydrological states at any spatial location within a structured mesh and it reproduces the hydrological response of contrasted catchments by taking advantage of spatially distributed meteorological forcings, physiographic data and hydrometric observations. Cost function gradient maps with respect to tunable parameters are a key feature of `smash` and can easily be combined to gradients of external operators such as a regionalization neural network (Huynh et al., 2024b) with chain rule in context of high dimensional optimization.

### 2.1   Forward model statement

Let $\Omega \subset \mathbb{R}^2$ denote a 2D spatial domain that can contain one to many gauges, with $x \in \Omega$ the spatial coordinate, $t \in \left]0, T\right]$ the physical time and $\mathcal{D}_\Omega$ a drainage plan over $\Omega$. The spatially distributed rainfall-runoff model $\mathcal{M}$ is a dynamic operator projecting the input fields of atmospheric forcings $\mathcal{I}$ onto the fields of surface discharge $Q$, internal states $\boldsymbol{h}$, and internal fluxes $\boldsymbol{q}$, as expressed in Equation 1.

$$\boldsymbol{U}(x,t) = [Q, \boldsymbol{h}, \boldsymbol{q}](x,t) = \mathcal{M}\left(\mathcal{D}_\Omega; \mathcal{I}(x,t); [\boldsymbol{\theta}, \boldsymbol{h}_0](x)\right) \tag{1}$$

with $\boldsymbol{U}(x,t)$ the modeled state-flux variables, $\boldsymbol{\theta}$ and $\boldsymbol{h}_0$ the spatially distributed parameters and initial states of the hydrological model.

The spatially distributed rainfall-runoff model $\mathcal{M}$ is obtained by partial composition (each operator taking various other input data and parameters) of the flow operators as follows:

$$\mathcal{M} = \mathcal{M}_{hy}(\,.\,,\mathcal{M}_{rr}(\,.\,,\mathcal{M}_{snw})) \tag{2}$$

Several process-based conceptual operators are available in `smash` for composing a model:

- **Snow operator** $\mathcal{M}_{snw}$ (optional): simulates melt flux $m_{lt}(x,t)$ feeding the hydrological operator in addition to rain.

  - *zero*: no module
  - *ssn*: degree-day module

- **Hydrological operator** $\mathcal{M}_{rr}$: simulation at pixel scale of elementary runoff $q_t(x,t)$ feeding the routing operator.

  - *gr4*: Génie Rural (GR)-like module (Perrin et al., 2003; Mathevet, 2005)
  - *gr5*: GR-like module (Le Moine, 2008; Ficchì et al., 2019)
  - *grd*: GR-like module (Perrin et al., 2003; Jay-Allemand et al., 2020)
  - *loieau*: GR-like module (Perrin et al., 2003; Folton and Arnaud, 2020)
  - *vic3l*: Variable Infiltration Capacity (VIC)-like module adapted from (Liang et al., 1994)

- **Routing operator** $\mathcal{M}_{hy}$: runoff routing from pixels to pixel to obtain spatio-temporal discharge $Q(x,t)$.

  - *lag0*: Instantaneous module
  - *lr*: Linear reservoir module
  - *kw*: Kinematic wave module. A classical 1D conceptual kinematic wave model, applied over a D8 drainage plan $\mathcal{D}_\Omega$ without channel. The model is solved numerically using a linearized implicit scheme (Chow et al., 1998).

The operators chaining principle is schematized in Figure 1 with input data and internal states and fluxes. The operators available in `smash` are listed above, and further detailed in Appendix D and in the online documentation (https://smash.recover.inrae.fr/math_num_documentation/forward_structure.html).

Originally, a differentiable descriptors-to-parameters mapping $\phi$ can be used to constrain spatially distributed conceptual parameters $\boldsymbol{\theta}(x)$ and initial states $\boldsymbol{h}_0(x)$ from physical descriptors $\boldsymbol{D}(x)$, for regionalization learning (Huynh et al., 2024b):

$$[\boldsymbol{\theta}, \boldsymbol{h}_0](x) = \phi(\boldsymbol{D}(x), \boldsymbol{\rho}), \forall x \in \Omega \tag{3}$$

with $\boldsymbol{D}$ the $N_D$-dimensional vector of physical descriptor maps covering the spatial domain $\Omega$ and $\boldsymbol{\rho}$ the vector of tunable regionalization parameters of the available mappings (written for $\boldsymbol{\theta}$ only for brevity):

1. A set $\mathcal{P}$ of multiple regression operators for each parameter of the forward hydrological model $\mathcal{M}$:

$$\theta_k(x, \boldsymbol{D}, \boldsymbol{\rho}_k) = s_k \left( \alpha_{k,0} + \sum_{d=1}^{N_D} \alpha_{k,d} D_d^{\beta_{k,d}}(x) \right), \forall k \in [1..N_\theta] \tag{4}$$

with $s_k(z) = l_k + (u_k - l_k)/(1 + e^{-z}), \forall z \in \mathbb{R}$ a transformation based on a sigmoid function with values in $[l_k, u_k]$ imposing constrains into the forward model such that $l_k < \theta_k(x) < u_k, \forall x \in \Omega$. The bounds $l_k$ and $u_k$ associated to each conceptual parameter $\theta_k$ are spatially uniform. The regional parameter control vector to estimate in this case is: $\boldsymbol{\rho} \equiv \left[ (\boldsymbol{\rho}_k)_{k=1}^{N_\theta} \right]^T \equiv \left[ \left( \alpha_{k,0}, (\alpha_{k,d}, \beta_{k,d})_{d=1}^{N_D} \right)_{k=1}^{N_\theta} \right]^T$; a multiple linear regression mapping is obtained by imposing $\beta_{k,d} = 1$.

2. An ANN denoted $\mathcal{N}$, consisting of a multilayer perceptron, aimed at learning the descriptors-to-parameters mapping such that:

$$\boldsymbol{\theta}(x, \boldsymbol{D}, \boldsymbol{\rho}) = \mathcal{N}(\boldsymbol{D}(x), \boldsymbol{W}, \boldsymbol{b}), \forall x \in \Omega \tag{5}$$

where $\boldsymbol{W}$ and $\boldsymbol{b}$ are respectively weights and biases of the neural network composed of $N_L$ dense layers. The architecture of the neural network and the forward propagation is detailed in Huynh et al. (2024b). Note that an output layer consisting of a scaling transformation is used to impose bound constraints as above. The regional control vector in this case is $\boldsymbol{\rho} \equiv [\boldsymbol{W}, \boldsymbol{b}]^T \equiv \left[ (\boldsymbol{W}_j, \boldsymbol{b}_j)_{j=1}^{N_L} \right]^T$

Note that:

- The available mappings for $\phi$ are also implemented to predict initial state vector $\boldsymbol{h}_0$ using physical descriptors fields that can include previous states and can be used for short range data assimilation (not studied here).

- By construction, the complete forward model $\mathcal{M}$ is learnable in terms of parameters regionalization, through the regionalization mapping $\phi$ embedded into $\mathcal{M}$ that is also differentiable, and its parameters $\boldsymbol{\rho}$ can be trained using cost gradient as explained after.

## 2.2 Inverse algorithm

Given observed and simulated discharge times series $\boldsymbol{Q}^* = (Q_{g=1..N_G}^*)^T$ and $\boldsymbol{Q} = (Q_{g=1..N_G})^T$ with $N_G$ the number of gauges over the study domain $\Omega$, the model misfit to multi-site observations is measured through a cost function $J$ that writes as

$$J(\boldsymbol{Q}^*, \boldsymbol{Q}) = \sum_{g=1}^{N_G} w_g j_g \left( Q_g^*, Q_g \right) + j_{reg}, \tag{6}$$

with $w_g$ the weight associated to the cost function $j_g$ at each gauge $g$, where $\sum_{g=1}^{N_g} w_g = 1$. This multi-gauge observation cost function is also used for mono-gauge calibration with $N_G = 1$. A regularization term $j_{reg}$ can be considered for ill-posed inverse problems (cf. Jay-Allemand et al. (2020, 2024)). In this study, equal weights were assigned to each gauge (i.e.

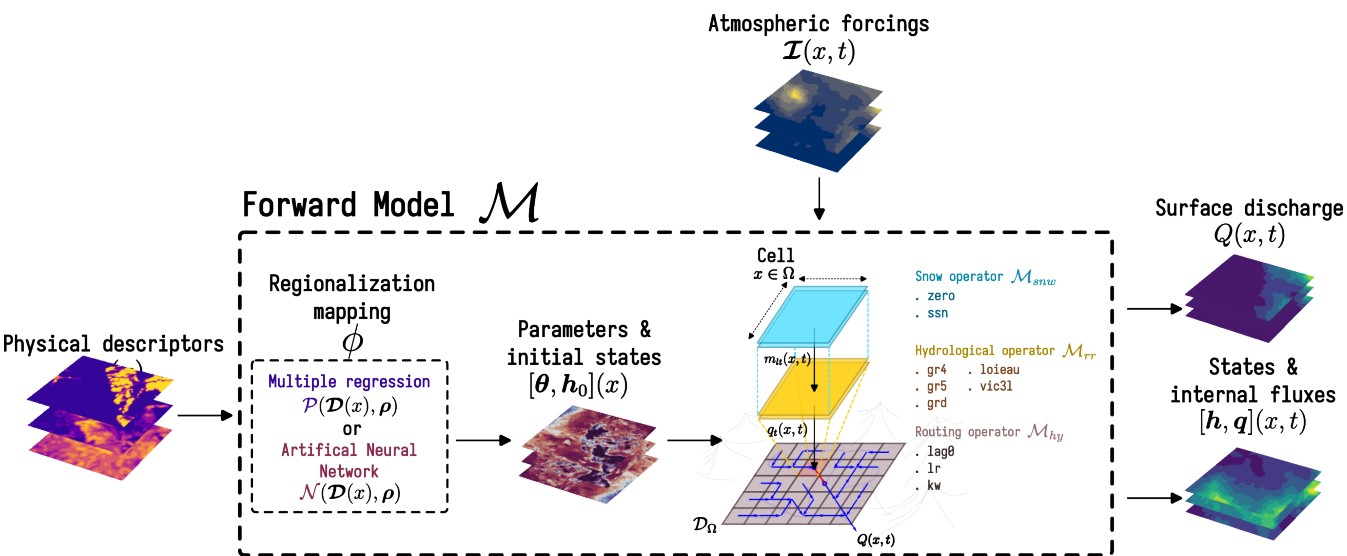

**Figure 1.** Flowchart of input data, operators chaining to obtain the forward differentiable model $\mathcal{M}$ that includes a learnable regionalization mapping $\phi$ (Huynh et al., 2024b), and simulated states and fluxes. The forward model $\mathcal{M}$ is obtained by partial composition (each operator taking various other input data and parameters) of the flow operators $\mathcal{M} = \mathcal{M}_{hy}(\,.\,,\mathcal{M}_{rr}(\,.\,,\mathcal{M}_{snw}))$.

$w_g = 1/N_G$), which corresponds to minimizing the average of the individual cost functions. Additionally, no regularization term was applied.

The gauge cost function is defined as

$$j_g(Q_g^*, Q_g) = \sum_{c=1}^{N_C} w_c j_c\left(Q_g^*, Q_g\right), \tag{7}$$

with $j_c$ being respectively based on any efficiency metric ($NSE$, $KGE$ (Gupta et al., 2009), etc) or a signature-based cost function including $N_C$ continuous and event-based components (Huynh et al., 2023) and $w_c$ their relative weights. For multi-score calibration strategy using continuous $NSE$ and event-based flood signatures see Huynh et al. (2023).

The global cost function $J$ is defined as a convex and differentiable function, involving the response of the forward model $\mathcal{M}$ through its output $Q$, and consequently depends on the model parameters $\boldsymbol{\theta}$, hence on the parameters $\boldsymbol{\rho}$ of the regional mapping $\phi$ when used (Eq. 3).

Therefore, the optimization problem formulates as in Equation 8:

$$\hat{\boldsymbol{\rho}} = \arg\min_{\boldsymbol{\rho}} J\left(\boldsymbol{Q}^*, \mathcal{M}(., \phi(., \boldsymbol{\rho}))\right) \tag{8}$$

This high-dimensional inverse problem can be tackled with gradient-based optimization algorithms. A limited-memory quasi-Newton approach, such as L-BFGS-B (Zhu et al., 1997), is suitable for smooth objective functions, while an adaptive learning rate approach, exemplified by Adam (Kingma and Ba, 2014), is effective for non-smooth objective functions. These approaches necessitate obtaining the gradient $\nabla_{\boldsymbol{\rho}} J$ of the cost function with respect to the tunable control parameter $\boldsymbol{\rho}$ obtained by solving

the adjoint $D_\rho\mathcal{M}$ of the forward model $\mathcal{M}$. The adjoint model is obtained by automatic differentiation using the `Tapenade` engine (Hascoet and Pascual, 2013). The complete forward model and VDA process are illustrated in Figure 2.

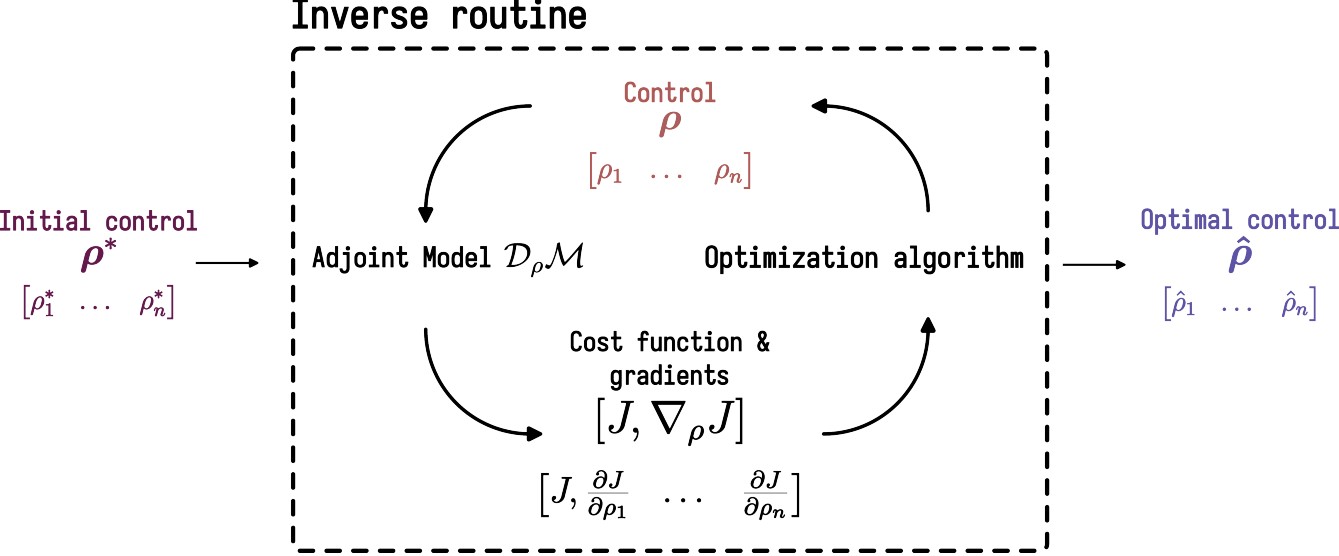

**Figure 2.** Flowchart of the inverse algorithm that uses $\nabla_\rho J$ the cost gradient with respect to the tunable control parameter $\rho$ obtained by solving the adjoint model $D_\rho\mathcal{M}$ of the forward model $\mathcal{M}$, obtained by automatic source code differentiation and enabling accurate gradient computation (adapted from VDA course Monnier (2024)).

## 3 Computational software and performance

In this section, we focus on code architecture, documentation and computational performance. `smash` is based on a computationally efficient Fortran core enabling parallel computations over large domains with `OpenMP` (Dagum and Menon, 1998), and that is automatically differentiable with the `Tapenade` engine (Hascoet and Pascual, 2013) to generate the numerical adjoint model. It is interfaced in Python using `f90wrap` (Kermode, 2020) to provide a user-friendly and versatile interface for quick learning and efficient development, as well as to directly make accessible the wealth of Python modules and libraries (Tab. 1) developed by a large and active community (Data pre/post-Processing, Geographic Information System, Deep Learning, etc).

### 3.1 From sources to ready-to-use Python library

`smash` contains a Python core for all the user interface functions, both pre- and post-processing, and a Fortran core (with a few C files) for high-performance numerical computations. In order to produce a Python library, including binary files, that can be installed directly from the package manager, `PyPI`, several steps are necessary. The first step is to generate the Fortran adjoint file from the Fortran sources. This is done via `Tapenade` automatic differentiation engine (Hascoet and Pascual, 2013), which requires the use of Java. Next, the Fortran code is wrapped for use in Python. `f90wrap` (Kermode, 2020) builds

**Table 1.** External Python libraries used by smash.

| Library | Website | Reference | Description |
|---------|---------|-----------|-------------|
| NumPy | https://numpy.org | Harris et al. (2020) | Numerical computing |
| SciPy | https://scipy.org | Virtanen et al. (2020) | |
| pandas | https://pandas.pydata.org | pandas development team (2020) | Data analysis and manipulation tool |
| f90wrap | https://github.com/jameskermode/f90wrap | Kermode (2020) | Fortran to Python interface generator |
| Rasterio | https://rasterio.readthedocs.io/en/stable | | Input/output |
| h5py | https://docs.h5py.org/en/stable | | |

on the capabilities of the popular F2PY utility by generating a simpler Fortran interface to the original Fortran sources which is then suitable for wrapping with F2PY, together with a higher-level Pythonic wrapper that makes the existence of an additional layer transparent to the final user. The entire build system (except for the generation of the adjoint file, which is external for debugging reasons) is handled by meson, a multi-platform, multi-language open-source build system that allows us to generate smash binaries on Linux, macOS and Windows quite easily (Fig. 3).

## 3.2 Documentation

The smash online documentation (Fig. 4) is divided into four main sections:

– Getting Started (https://smash.recover.inrae.fr/getting_started)

This section describes how to install smash from the Python package index PyPI.

– User Guide (https://smash.recover.inrae.fr/user_guide)

This section provides step-by-step examples (and scripts) from basic (simulation run) to complex (regionalization) applications of smash as well as input data conventions.

– API Reference (https://smash.recover.inrae.fr/api_reference)

This section details the different modules and the application programming interface. Modules are documented using the NumPy-style Python docstring.

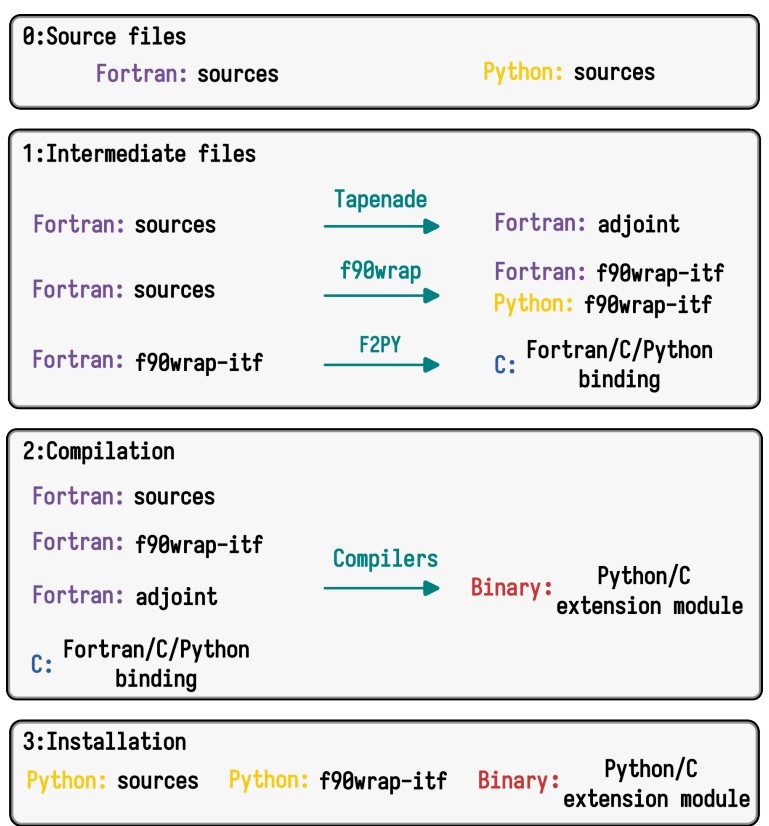

**Figure 3.** The `smash` build system framework. It starts with source files written in Fortran and Python (0). In the intermediate step (1), Fortran sources are processed by `Tapenade` to generate the adjoint code, and wrapped using `f90wrap` to create Python interfaces (f90wrap-itf). The `F2PY` tool is then used to generate a Fortran/C/Python binding from the wrapped interfaces. During the compilation step (2), the original Fortran sources, adjoint code, and f90wrap-itf are compiled with appropriate compilers to produce a binary Python/C extension module. Finally, in the installation step (3), the Python module is assembled, combining the original Python sources with the f90wrap-itf and the compiled binary Python/C extension module, making the high-performance Fortran code accessible from Python.

 – Math / Num Documentation (https://smash.recover.inrae.fr/math_num_documentation)

This last section details the conceptual and mathematical basis of the forward and inverse modeling problems, their
numerical resolution along with optimization and estimation algorithms.

The whole documentation is implemented using `Sphinx` to automatically compile and update an online version.

### 3.3 Computational performance

In this section, we compare the performance of `smash` in terms of computation time and memory usage between direct and adjoint runs (an adjoint run is equivalent to a single call to the adjoint model $D_\rho \mathcal{M}$ here). The aim is to highlight the resources
required to run `smash` on configurations similar to real cases. We compare `smash` over 3 zones: Sardinia, Great Britain/Ireland

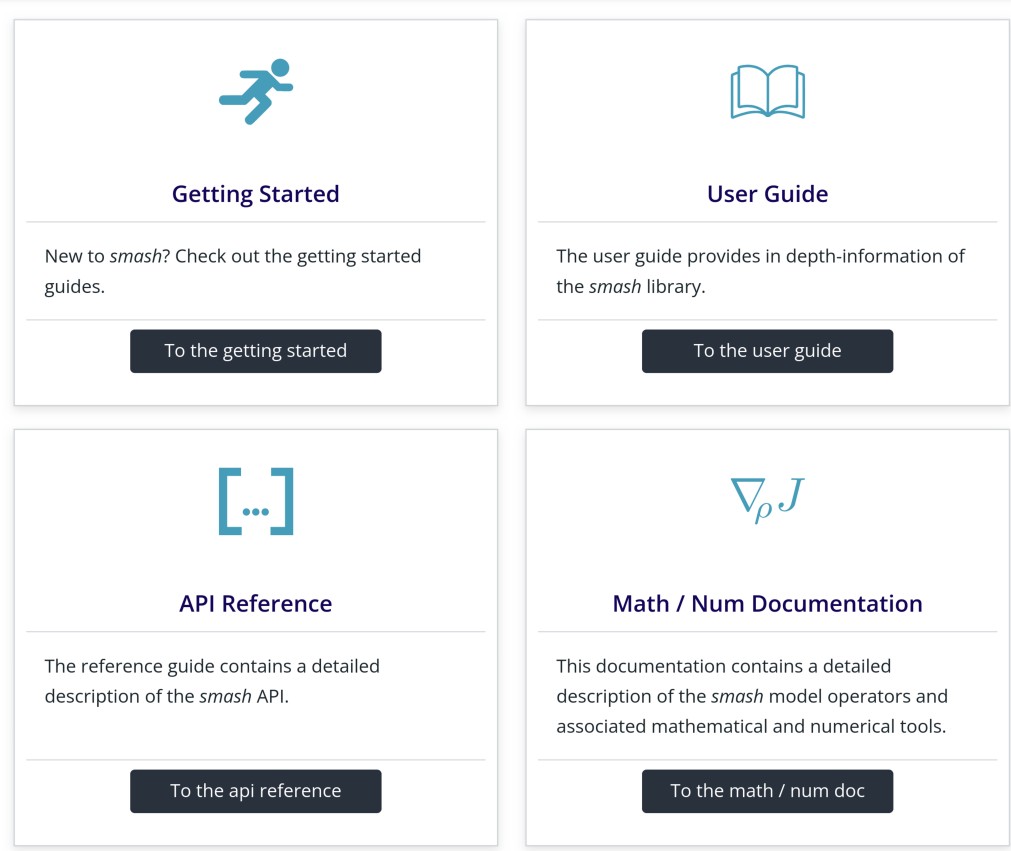

**Figure 4.** `smash` documentation home page accessible at https://smash.recover.inrae.fr.

and North America at a spatial resolution of 1'30" ($\sim$ 3km $\times$ 3km) over a period of 1 year, from July 31, 2010 to July 31, 2011, randomly chosen, at a daily time step. These 3 zones were chosen simply to provide 3 zones of variable surface area (Fig. 5). In addition to the 3 zones, with `smash` enabling different assemblies of operators, two structures are compared, s1 and s2, representing respectively the simplest ($\mathcal{M}_{snw}$: *zero*, $\mathcal{M}_{rr}$: *grd* and $\mathcal{M}_{hy}$: *lag0*) and the most complex ($\mathcal{M}_{snw}$: *ssn*,

$\mathcal{M}_{rr}$: *vic3l* and $\mathcal{M}_{hy}$: *kw*) structure in terms of number of operations per cell. All the simulations (1 year of simulation at daily time step) were run on a server with AMD EPYC 7643 CPUs (Annexe E) and 255 GB of RAM.

The range of computation times across all simulations (Fig. 6) varies from approximately 0.1 second for a direct run with 8 threads on the Sardinia region using the s1 structure to just over an hour for an adjoint run with 1 thread on the North America region using the s2 structure. Systematically, regardless of the region, number of threads, or type of run, the difference in

computation time between the s1 and s2 structures is about a factor of 2. Regarding the differences between a direct run and an adjoint run, the computation time factor varies depending on the number of threads, ranging from a factor of 12 for 1 thread

to a factor of 6 for 16 threads. This difference highlights better thread scaling for the adjoint run, with a speedup of around 4 for a direct run and 7 for an adjoint run with 16 threads likely because the adjoint run is more computationally demanding than the forward run. It is worth noting that in the case of the Sardinia region, which has the fewest grid cells, thread scaling is poor
compared to the other two regions, even reaching the limit where thread overhead increases the computation time. Although the time-stepping loop cannot be parallelized, and the routing scheme in `smash` must be solved sequentially from upstream to downstream, allowing only partial parallelization over the entire spatial domain, the approach still offers a substantial reduction in computation time.

About memory usage (Tab. 2), values range from 0.17 GB for a direct run on the Sardinia region with the s1 structure to 27
GB for an adjoint run on the North America region with the s2 structure. Systematically, memory usage is higher in an adjoint run than in a direct run and scales with the size of the domain. The main contributor to memory usage in an adjoint run is the forward sweep, which includes a time-stepping loop where iteration $n$ depends on the results of previous iterations. The memory allocation during the forward sweep is freed during the backward sweep, but it still results in a significant memory peak. This memory peak has been considerably reduced in the `smash` version presented here by including checkpoints within
the time-stepping loop. These checkpoints allow to alternate between forward and backward sweeps, leading to much smaller memory peaks compared to a single sweep. The downside of using checkpoints is the increase in computation time, but this was considered less significant compared to the memory savings (see Hascoet and Pascual (2013) for further details about forward/backward sweep and checkpointing).

In conclusion, these computation times and memory usage demonstrate the feasibility of the model for large-scale appli-
cations. The critical point is parameters estimation. In the case of parameters estimation using a gradient-based optimizer, one or more adjoint runs are evaluated at each iteration, significantly multiplying the total computation time. As an example, Huynh et al. (2024b) performed a calibration at a spatial resolution of 1km over a domain of more than 20,000 cells, and at a temporal resolution of one hour over a period of 4 years. The calibration required 350 calls to the adjoint model, resulting in a computation time of around 180 hours. Currently, memory usage in an adjoint run is less of a limiting factor than computation
time for large-domain applications. Thus, further improving computation time is a priority to expand the model's application to finer spatial and temporal scales.

**Table 2.** Memory usage in Gigabyte (GB) for both direct and adjoint run simulations over a period of 1 year at daily time step.

| Zone | Structure | Memory Usage (GB) Direct run | Memory Usage (GB) Adjoint run |
|---|---|---|---|
| Sardinia | s1 | 0.17 | 0.18 |
| | s2 | 0.18 | 0.20 |
| Great Britain/Ireland | s1 | 0.37 | 0.46 |
| | s2 | 0.52 | 0.86 |
| North America | s1 | 7.64 | 11.27 |
| | s2 | 12.7 | 26.68 |

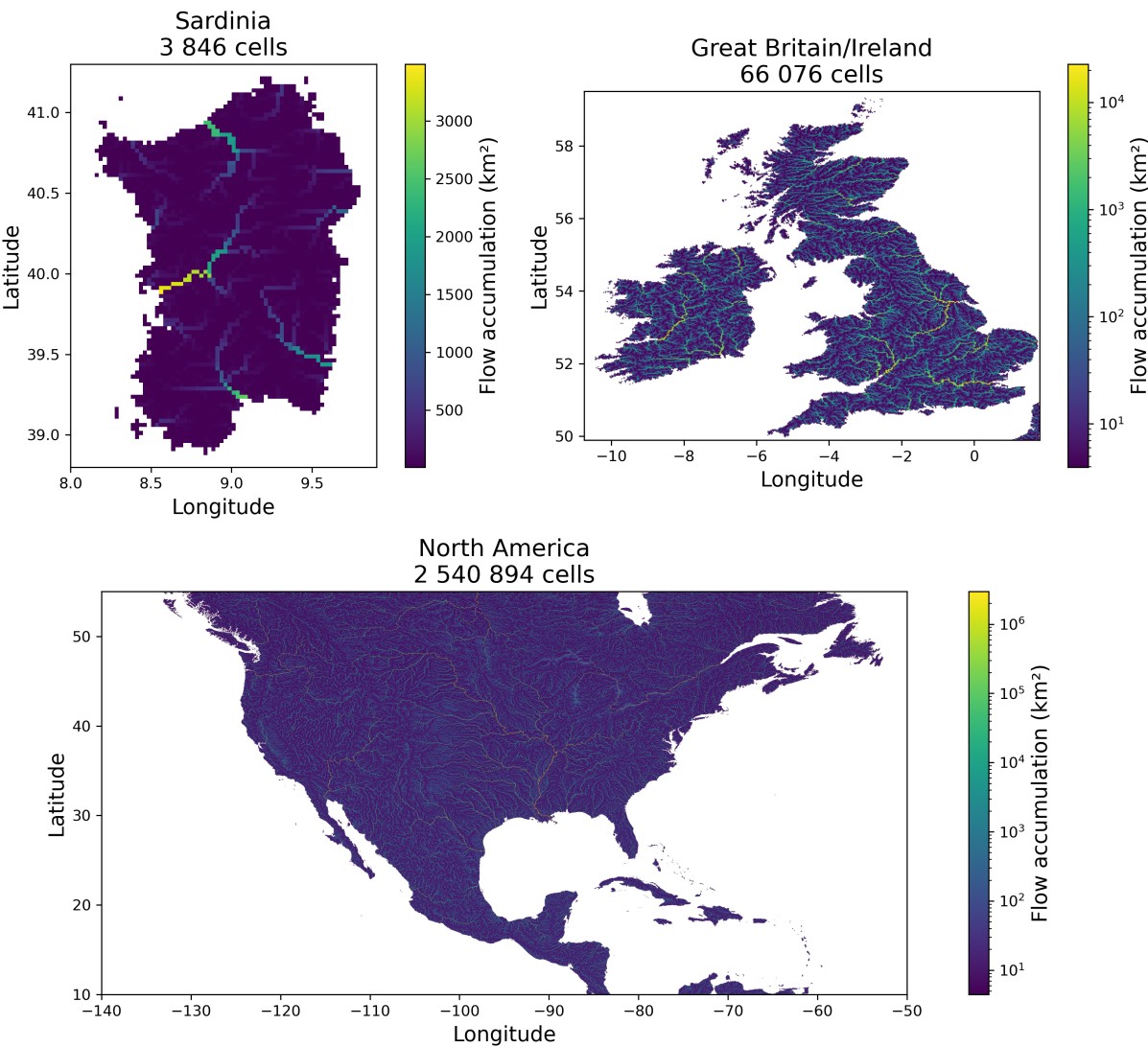

**Figure 5.** Spatial representations of the three different geographical regions used in the performance benchmarks.

## 4   Applications

### 4.1   Numerical experiments presented

Main functionalities and operators of `smash` are illustrated on open-source global datasets over contiguous United States
(CONUS) (Tab. 3, Fig. 7) and on a higher resolution open-source regional dataset in France (Tab. 4, Fig. 7). Models on
CONUS will be at a spatial resolution of 1'30" ($\sim$ 3km $\times$ 3km) and daily time step while higher resolution models will be
setup on the French case at 500-m spatial resolution and hourly time step. The numerical results presented are:

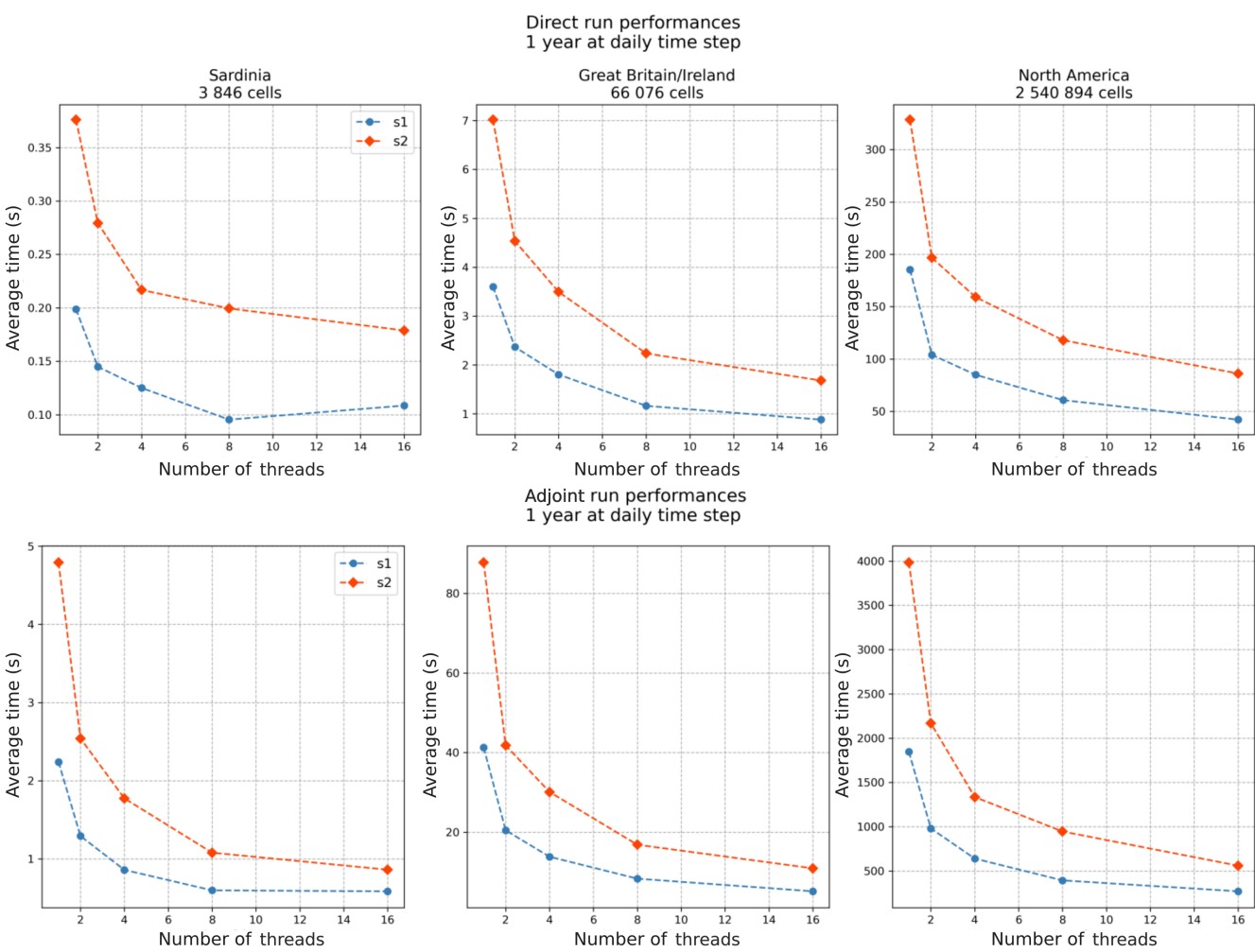

**Figure 6.** Benchmarking results for both direct (top row) and adjoint (bottom row) run simulations over a period of 1 year at daily time step, using varying numbers of threads (from 1 to 16). Each plot corresponds to a different geographical region: Sardinia, Great Britain/Ireland and North America from left to right.

– Split-sample temporal cross-validation of different model structures combinations over CONUS (Sect. 4.2)

– Regionalization over CONUS (Sect. 4.3)

– High-resolution regionalization over the Aude river in France (Sect. 4.4).

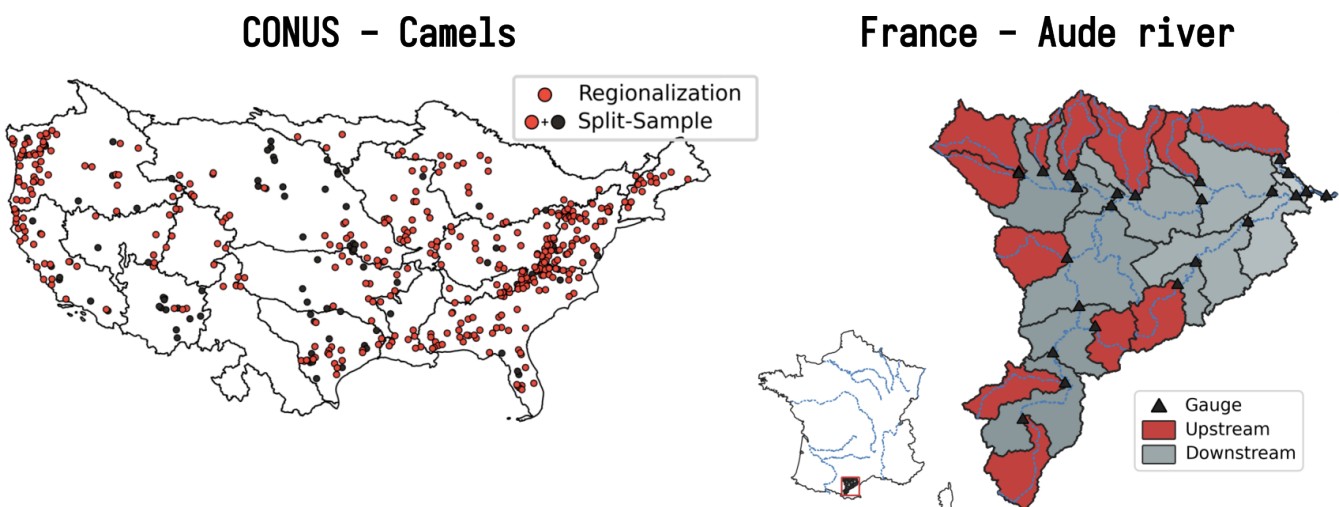

**Figure 7.** Location of the catchments for the CONUS (left) and France (right) applications. For the split-sample test over CONUS, all the 482 catchments from the CAMELS dataset (Addor et al., 2017) are used (orange and black circles) whereas for the regionalization a subset of 398 catchments is used (only orange circles), removing catchments whose performance are less than 0.75 $KGE$ from local calibration. For the France application over the Aude river, a set of 25 sub-catchments is used for regionalization with 12 upstream catchments (red shaded regions) and 13 downstream catchments (gray shaded regions).

## 4.2   CONUS - CAMELS - Split-sample temporal cross-validation

### 4.2.1   Numerical experiment settings

A set of 482 catchments (Fig. 7) is modeled with the following experimental design:

– A set of hydrological models is considered, including 4 GR-like structures (Perrin et al., 2003) ($\mathcal{M}_{rr}$: *gr4*, *gr5*, *grd*, *loieau*) and one VIC-like structure (Liang et al., 1994) ($\mathcal{M}_{rr}$: *vic3l*). The way in which these models are integrated into `smash`, that differs from the original models, is described in the documentation (https://smash.recover.inrae.fr/math_num_documentation/forward_structure.html) in the forward structure section. For each hydrological model, the same snow module ($\mathcal{M}_{snw}$: *ssn*) and routing module ($\mathcal{M}_{hy}$: *kw*) are used. A description of the calibrated parameters is provided in the Appendix A.

**Table 3.** Model input data from open-source databases available worldwide used over CONUS: atmospheric forcings $\mathcal{I} = \{P, N, E, T\}$, flow direction map $\mathcal{D}_\Omega$, physical descriptors $D = \{d_1, ..., d_6\}$ for regionalization based on Beck et al. (2020) study and discharge time series $Q^*$. The liquid and solid precipitation, $P$ and $N$, derive from the Multi-Source Weighted-Ensemble Precipitation (MSWEP) (Beck et al., 2019) divided into liquid and solid parts using a parametric S-shaped curve (Garavaglia et al., 2017) and disaggregated from 0.1° to 0.025°. The temperature and potential evapotranspiration, $T$ and $E$, derive from ERA5 (Hersbach et al., 2020) disaggregated from 0.25° to 0.025°, using the Oudin formula (Oudin et al., 2005) to obtain the potential evapotranspiration. The flow direction, $\mathcal{D}_\Omega$, from MERIT Hydro IHU (Eilander et al., 2021) was upscaled from 0.008° to 0.025° using `pyflwdir` (Eilander, 2023). The topographic slope, $d_1$, derives from MERIT DEM (Yamazaki et al., 2017) upscaled from 0.008° to 0.025° using `gdaldem slope`. The sand and clay content, $d_2$ and $d_3$, from SoilGrids (Hengl et al., 2017) were upscaled and reprojected from 250m to 0.025°. The meteo-climatic data, $d_4$, $d_5$ and $d_6$ derive from $P$ and $E$. The discharge time series, $Q^*$, comes from Caravan-CAMELS (Kratzert et al., 2023; Addor et al., 2017).

| Notation | Type | Description | Unit | Source |
|---|---|---|---|---|
| $P$ | Atmospheric forcing | Liquid precipitation | $(mm/day)$ | MSWEP (Beck et al., 2019) |
| $N$ | Atmospheric forcing | Solid precipitation | $(mm/day)$ | MSWEP (Beck et al., 2019) |
| $E$ | Atmospheric forcing | Potential evapotranspiration using Oudin formula (Oudin et al., 2005) | $(mm/day)$ | ERA5 temperature (Hersbach et al., 2020) |
| $T$ | Atmospheric forcing | Temperature | $(°C)$ | ERA5 temperature (Hersbach et al., 2020) |
| $\mathcal{D}_\Omega$ | Topography | Flow direction | $(-)$ | MERIT Hydro IHU (Eilander et al., 2021) |
| $d_1$ (slope) | Topography | Topographic slope | $(°)$ | MERIT (Yamazaki et al., 2017) |
| $d_2$ (sand) | Soil | Sand content, averaged over all layers | $(g/kg)$ | SoilGrids (Hengl et al., 2017) |
| $d_3$ (clay) | Soil | Clay content, averaged over all layers | $(g/kg)$ | SoilGrids (Hengl et al., 2017) |
| $d_4$ (prcp) | Meteo-Climatic | Mean annual precipitation | $(mm/yr)$ | MSWEP (Beck et al., 2019) |
| $d_5$ (pet) | Meteo-Climatic | Mean annual potential evapotranspiration using Oudin (Oudin et al., 2005) | $(mm/yr)$ | ERA5 temperature (Hersbach et al., 2020) |
| $d_6$ (hi) | Meteo-Climatic | Mean annual humidity index (ratio of precipitation to potential evapotranspiration) | $(-)$ | MSWEP (Beck et al., 2019), ERA5 temperature (Hersbach et al., 2020) |
| $Q^*$ | Hydrometric | Discharge time series | $(m^3/s)$ | Caravan-CAMELS (Kratzert et al., 2023; Addor et al., 2017) |

– A split-sample temporal validation procedure (Klemeš, 1983) is set up splitting the time window covered by hydrometric data into two complementary subsets over sub-periods of 7 years: $p1$ (from 1 August 2000 to 31 July 2007) and $p2$ (from

**Table 4.** Model input data from national open-source databases used over the Aude river in France: atmospheric forcings $\mathcal{I} = \{\boldsymbol{P}, \boldsymbol{E}\}$, flow direction map $\mathcal{D}_\Omega$, physical descriptors $\boldsymbol{D} = \{d_1, ..., d_7\}$ for regionalization and discharge time series $\boldsymbol{Q^*}$. The liquid precipitation, $\boldsymbol{P}$, comes from the ANTILOPE J+1 Météo-France product (Champeaux et al., 2009), a radar-gauge reanalysis disaggregated from 1km to 500m. The potential evapotranspiration, $\boldsymbol{E}$, derives from the SAFRAN Météo-France temperature (Quintana-Seguí et al., 2008; Vidal et al., 2010) disaggregated from 8km to 500m, using the Oudin formula (Oudin et al., 2005) to obtain a daily interannual potential evapotranspiration. The flow direction, $\mathcal{D}_\Omega$, comes from HydroDem (Leblois and Sauquet, 1999). The land cover data, $d_1$, $d_2$, $d_3$ and $d_4$ derive from CORINE Land Cover 2018 (doi.org) rasterized at 50m and upscaled to 500m using the average resampling method. The topographic slope, $d_5$, derives from HydroDem DEM (Leblois and Sauquet, 1999) using `gdaldem slope`. The drainage density, $d_6$ comes from (Organde et al., 2013) representing the number of cells crossed by a river. The percentage of karst, $d_7$, comes from BDLISA (https://bdlisa.eaufrance.fr/) rasterized at 50m and upscaled to 500m using the average resampling method. The discharge time series, $\boldsymbol{Q^*}$, comes from HydroPortail Service Central Vigicrues (https://hydro.eaufrance.fr/).

| Notation | Type | Description | Unit | Source |
|---|---|---|---|---|
| $\boldsymbol{P}$ | Atmospheric forcing | Liquid precipitation | $(mm/h)$ | Antilope J+1 from Météo-France |
| | | | | (Champeaux et al., 2009) |
| $\boldsymbol{E}$ | Atmospheric forcing | Potential evapotranspiration using | $(mm/h)$ | SAFRAN temperature from |
| | | the Oudin formula (Oudin et al., | | Météo-France (Quintana-Seguí et al., |
| | | 2005) | | 2008; Vidal et al., 2010) |
| $\mathcal{D}_\Omega$ | Topography | Flow direction | $(-)$ | HydroDem (Leblois and Sauquet, 1999) |
| $d_1$ (*artif*) | Land cover | Artificial cover rate | $(-)$ | CORINE Land Cover 2018 (doi.org) |
| $d_2$ (*forest*) | Land cover | Forest cover rate | $(-)$ | CORINE Land Cover 2018 (doi.org) |
| $d_3$ (*veg*) | Land cover | Vegetation cover rate | $(-)$ | CORINE Land Cover 2018 (doi.org) |
| $d_4$ (*ow*) | Land cover | Open water cover rate | $(-)$ | CORINE Land Cover 2018 (doi.org) |
| $d_5$ (*slope*) | Topography | Topographic slope | $(°)$ | HydroDem (Leblois and Sauquet, 1999) |
| $d_6$ (*ddr*) | Topography | Drainage density | $(-)$ | (Organde et al., 2013) |
| $d_7$ (*karst*) | Hydrogeology | Percentage of karst | $(\%)$ | BDLISA (https://bdlisa.eaufrance.fr/) |
| $\boldsymbol{Q^*}$ | Hydrometric | Discharge time series | $(m^3/s)$ | HydroPortail SCHAPI |
| | | | | (https://hydro.eaufrance.fr/) |

1 August 2007 to 31 July 2014) are both used for calibration and validation. For each period, the 10 preceding years are used as model "warm-up".

– 2 calibration mappings on each catchment are tested:

    ◦ Uniform: spatially uniform parameters (gradient-free optimization)

    ◦ Distributed: spatially distributed parameters (gradient-based optimization)

– The use of a single-gauge cost function based on the $KGE$ ($J = 1 - KGE$)

### 4.2.2 Results

The performance of the models resulting from the spatially uniform or distributed calibration is evaluated using the Kling-Gupta Efficiency ($KGE$) for both the calibration and the validation on period $p2$ only for brevity in this software article (Fig. 8). Overall performance is satisfactory, with a median between 0.8 and 0.87 for $KGE$ over the calibration period and between 0.72 and 0.78 for $KGE$ over the validation period. With regard to the calibration method, for any model, calibration and validation performances are better with a spatially distributed calibration. This is an expected result for the calibration period, given that spatially distributed calibration is over-parameterized and offers the maximum level of flexibility in the search for the optimal set of parameters, unlike spatially uniform calibration, which is under-parameterized, imposing a single parameter set for each catchment. However, despite this over-parametrization with calibration of spatially distributed parameters, which can lead to over-fitting over the calibration period, the models offer good performance in temporal validation. The differences between the structures are mainly explained by (i) the varying levels of model complexity, 2 parameters for the *grd* model and 4 for the *gr5* model, and (ii) the expert knowledge of the different models, which influences, among other things, the choice of initial values, bounds and parameters to be optimized. The `smash` historical development based on the GR-like models led to a much more substantial expert knowledge than for the VIC-like model recently implemented. Summary statistics of the calibrated parameters are provided in Appendix A.

Concerning the spatial distribution of $KGE$ values (Fig. 9), the results for the *gr4* hydrological model after a spatially distributed calibration show that the best performances are located over the east and west sides of CONUS while worst performances are located over the Great Plains area. This spatial pattern of hydrological model performance has also been obtained in other studies (Newman et al., 2015; Beck et al., 2016; Mizukami et al., 2017).

### 4.3 CONUS - CAMELS - Regionalization

### 4.3.1 Numerical experiment settings

A set of 398 catchments (Fig. 7) from the CAMELS dataset (Addor et al., 2017) is evaluated in a regionalization context at a spatial resolution of 1'30" and at a daily time step using worldwide databases (Tab. 3). The experimental design is as follows:

– Selection of a subset of catchments from Section 4.2, eliminating catchments where $KGE < 0.75$ from local calibration. This selection is made in order to avoid introducing catchments whose performance could greatly degrade the calibration metric in a multi-gauge context.

– One hydrological model is considered, identical to the *gr4* model in Section 4.2 with the same snow and routing module ($\mathcal{M}_{snw}$: *ssn*, $\mathcal{M}_{rr}$: *gr4*, $\mathcal{M}_{hy}$: *kw*). A description of the calibrated parameters is provided in the Appendix B.

– A spatio-temporal validation procedure is set up by:

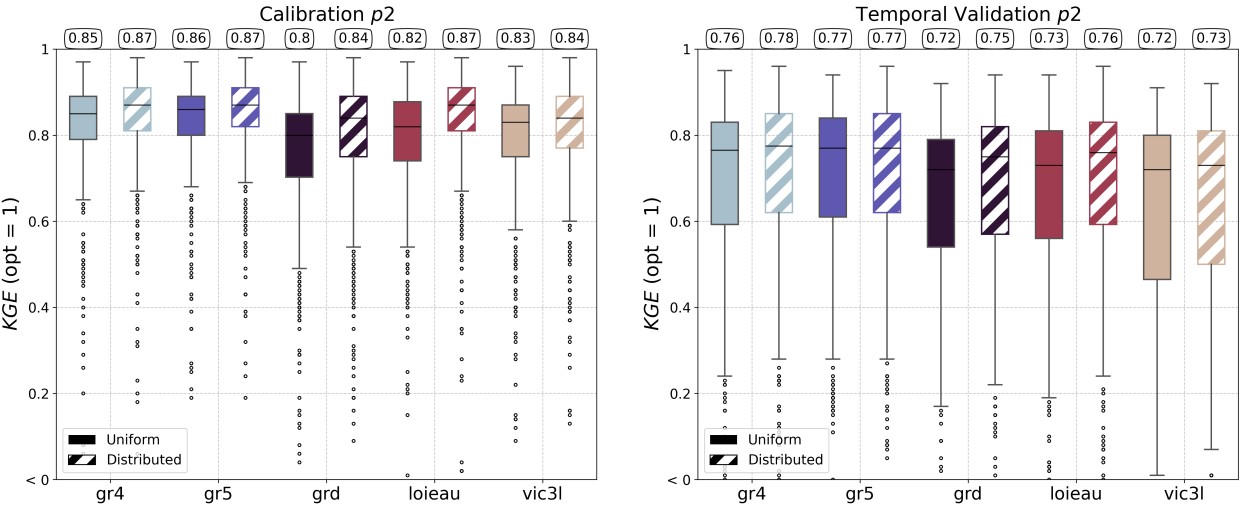

**Figure 8.** Comparison of the Kling-Gupta Efficiency ($KGE$) performance of different `smash` hydrological models under spatially uniform and distributed calibration. The models evaluated include *gr4*, *gr5*, *grd*, *loieau*, and *vic3l*. The left panels show results for the calibration, while the right panels display results for temporal validation on period *p2*. For each model, results are shown for spatially uniform (solid boxes) and spatially distributed (hatched boxes) calibrations with the median value highlighted at the top of the boxplot.

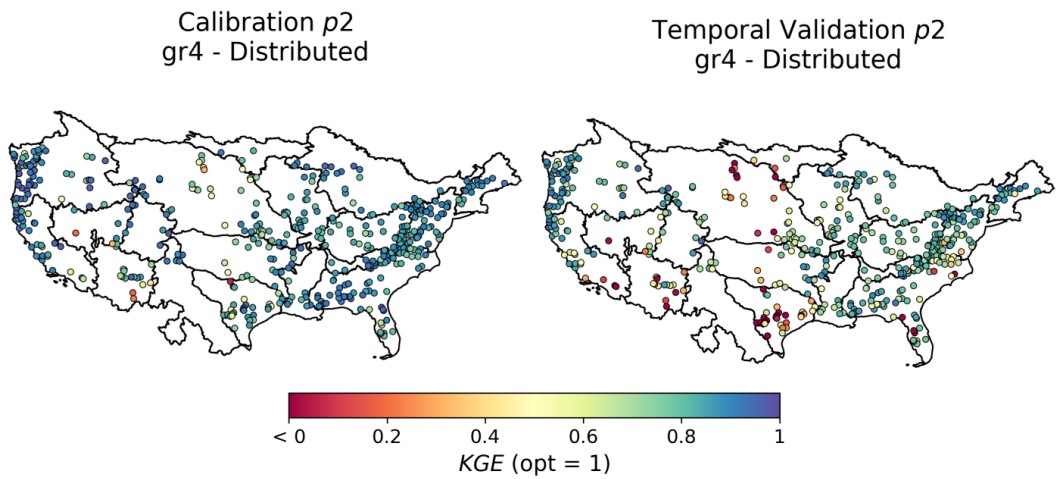

**Figure 9.** Spatial distribution of the Kling-Gupta Efficiency ($KGE$) scores across different catchments for the *gr4* model under local calibration with spatially distributed parameters. The left map shows $KGE$ scores during the calibration period while the right map shows results for temporal validation, using the *p2* period.

- ○ splitting the time window covered by hydrometric data into two complementary subsets over sub-periods of 7 years: *p1* (from 1 August 2000 to 31 July 2007) and *p2* (from 1 August 2007 to 31 July 2014), with *p1* used as calibration period and *p2* as validation period. For each period, 10 years are used as model "warm-up".

          ○ randomly splitting the catchment set into 4 groups, calibrating on 3 of the groups, with the 4th group held out and used for validation, then rotating such that each group is used for validation once.

    – 3 calibration mappings across the whole CONUS are tested:

        ○ Uniform: spatially uniform parameters (gradient-free optimization)

        ○ Multi-Linear: a multiple linear regression is used as transfer function from descriptors to spatialized parameters

          (gradient-based optimization)

        ○ ANN: a multi-layer perceptron composed of 3 hidden layers is used as a transfer function from descriptors to spatialized parameters (gradient-based optimization)

    – The use of a multi-gauge cost function based on the average $KGE$ of the calibrated catchments ($J = \frac{1}{N_G} \sum^{N_G} 1 - KGE$)

    – A final calibration over the total period $p1 + p2$ including all gauges with the ANN mapping and the same multi-gauge cost function to analyze the output model parameters and their correlations with input descriptors.

### 4.3.2 Results

The regional calibration over CAMELS dataset was performed on 4 groups of randomly selected catchments as explained above. The performances in spatial and/or temporal validation are shown in Figure 10 and detailed by catchment groups in Table B2 for the ANN mapping, which is the best performer. In spatio-temporal validation, the most challenging extrapolation case, a uniform mapping leads to a median $KGE$ of 0.5 while the two regionalization methods result in $KGE$ of 0.61 or 0.63, respectively for Multi-Linear and ANN mapping. These fairly good performances, obtained with a relatively simple setup in terms of descriptors and cost function in particular, are comparable with regionalization works in the literature (Mizukami et al., 2019; Beck et al., 2020; Feng et al., 2024). In a similar way to the previous section (Sect. 4.2), the worst performances are located in the Great Plains but also, more clearly than in the local calibrations, in the western part of the country.

Following the evaluation of performance in spatio-temporal validation, a regional calibration with the ANN mapping over the period including $p1$ and $p2$ and with all gauges is carried out. This calibration enables us to analyze the correlations between the physiographic descriptors and the parameters obtained (Fig. 11), in a more robust way than with the various spatio-temporal validation groups. The correlation matrix highlights significant linear correlations, notably between the melt coefficient ($k_{mlt}$) and the topographical slope ($d_1$), the size of the production reservoir ($c_p$) and the mean annual rainfall ($d_4$), as well as the moisture content ($d_6$) and the routing parameters ($a_{kw}$, $b_{kw}$) with the same moisture content ($d_6$). Conversely, the exchange parameter ($k_{exc}$), a parameter directly affecting the model's mass balance in a non-conservative way, shows almost no linear correlation with the descriptors and is almost spatially uniform over the whole domain around the value of 0. While a detailed regionalization study on CAMELS datasets using our original adjoint-based algorithms is beyond the scope of this software article, the achieved performance across this large sample already showcases the algorithm's potential for global applicability. It also demonstrates the algorithm's effectiveness in enforcing spatially distributed hydrologic model constraints at the pixel

scale, leading to seamless parameter maps at a reasonable computational cost. Regarding computation times, the calibration with ANN mapping over periods $p1$ and $p2$ took 95 hours. This calibration involved 350 iterations, corresponding to 350 calls to the adjoint model, and was performed using 16 threads. For comparison, a single adjoint model run takes approximately 16 minutes, whereas a direct model run takes around 5 minutes using the same number of threads.

Finally, leveraging the fully distributed nature of `smash`, regional streamflow maps can be generated. An example is shown in Figure 12, which illustrates the dynamics of Hurricane Katrina over a 6-day period from August 27 to September 1, 2005. Notably, the routing model used in this exercise, the kinematic wave, was applied uniformly across the entire domain, including areas outside its validity range, such as downstream of major rivers on flat topography. Further work focuses on enriching `smash` with hydraulic models, starting with 1D and 2D dynamic wave model that neglects the convective acceleration term, but retains local acceleration and pressure gradient terms (Bates et al., 2010) for numerical implementation simplicity. Additionally, physics-based differential equations for hydrologic water balance at the pixel scale will be incorporated. Hybrid physics-AI formulations, which embed neural networks capable of learning parametrization and potentially uncertain model operators from data, can be explored thanks to the differentiable nature of the models within `smash`.

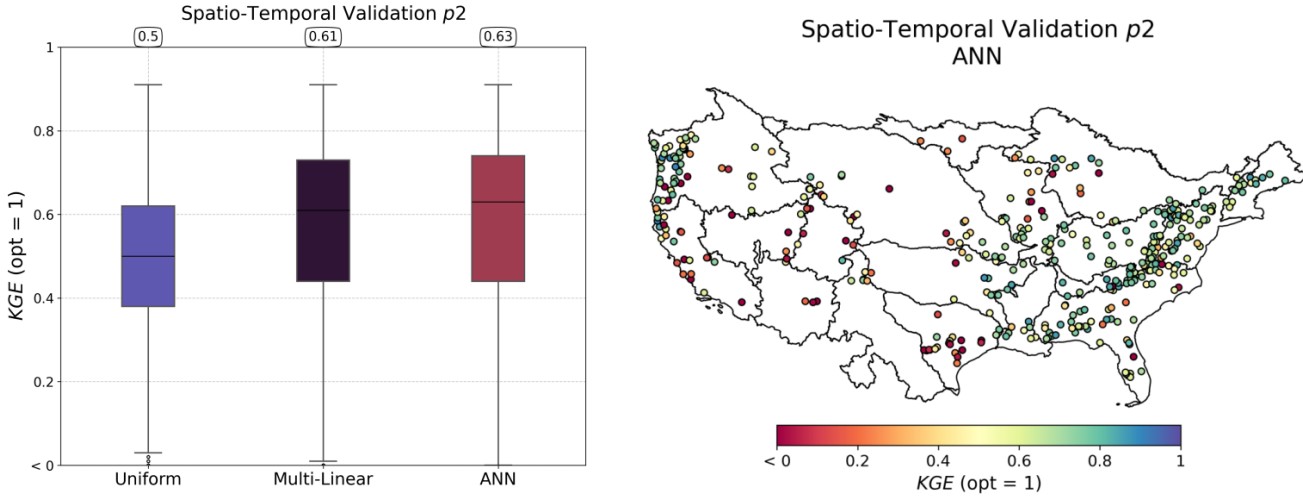

**Figure 10.** Spatio-temporal validation performance over period $p2$. The boxplots on the left panel represent the distribution of Kling-Gupta Efficiency ($KGE$) scores for three calibration methods: Uniform, Multi-Linear, and Artificial Neural Network (ANN). Median values are displayed at the top of each boxplot. The map on the right illustrates the spatial distribution of the $KGE$ values for the ANN mapping across different catchments.

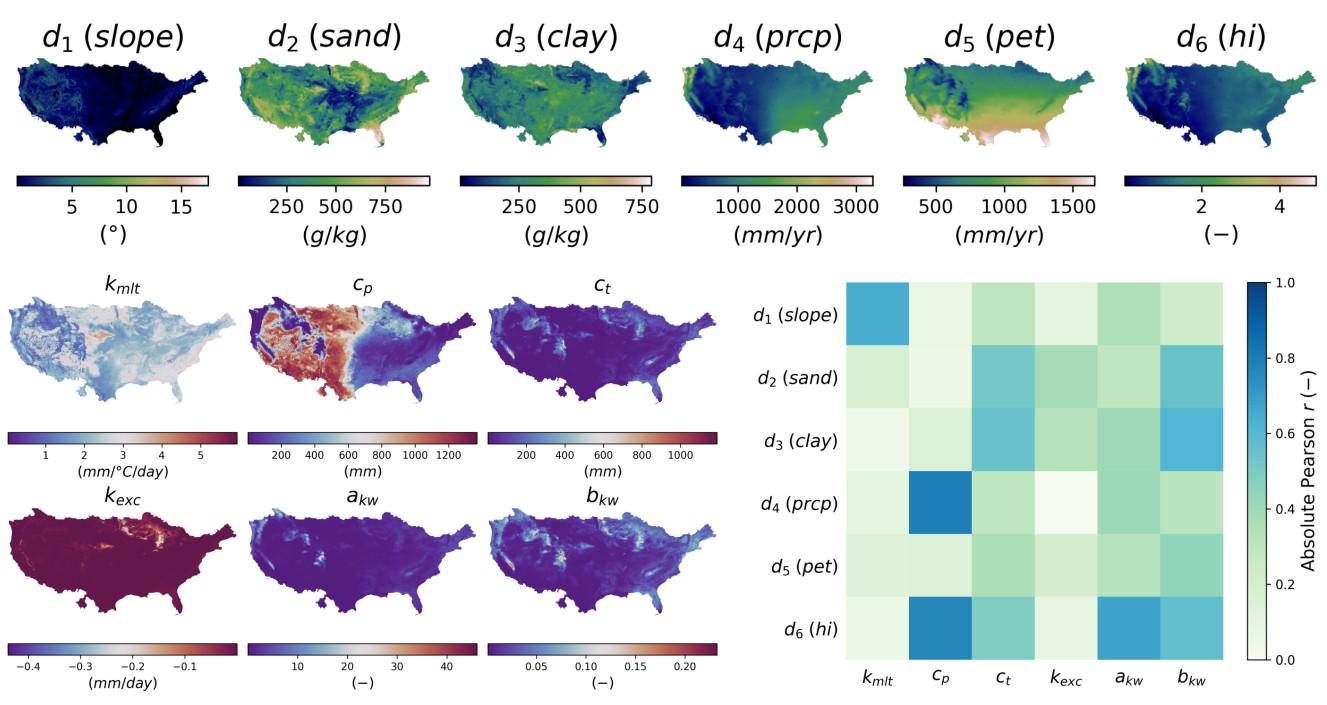

**Figure 11.** Analysis of input descriptors and output model parameters for the ANN mapping. Spatial distribution of physical descriptors ($d_1$-$d_6$) on the top panel, details provided in (Tab. 3); spatial distribution of calibrated hydrological parameters ($k_{mlt}, c_p, c_t, k_{exc}, a_{kw}, b_{kw}$) on the lower left panel and linear correlation between descriptors and parameters on the lower right panel.

### 4.4 France - Aude river - High-resolution regionalization

#### 4.4.1 Numerical experiment settings

A set of 35 catchments (Fig. 7) over the Aude river in France (Addor et al., 2017) is evaluated in a regionalization context at a spatial resolution of 500m and at a hourly time step using national databases (Tab. 4). This section is similar to the previous one, using the same regionalization method, but with differences in the gauges selected for calibration and validation, as well as in the cost function. This section focuses on national data at a finer spatio-temporal scale, applying the method at the watershed level, which is more relevant for operational flood forecasting. The experimental design is as follows:

– One hydrological model is considered, identical to the *gr4* model in Section 4.2 with the same routing module but without any snow modeling given the limited impact of snow in this Mediterranean basin ($\mathcal{M}_{snw}$: *zero*, $\mathcal{M}_{rr}$: *gr4*, $\mathcal{M}_{hy}$: *kw*). A description of the calibrated parameters is provided in the Appendix C

– A spatio-temporal validation procedure is set up by:

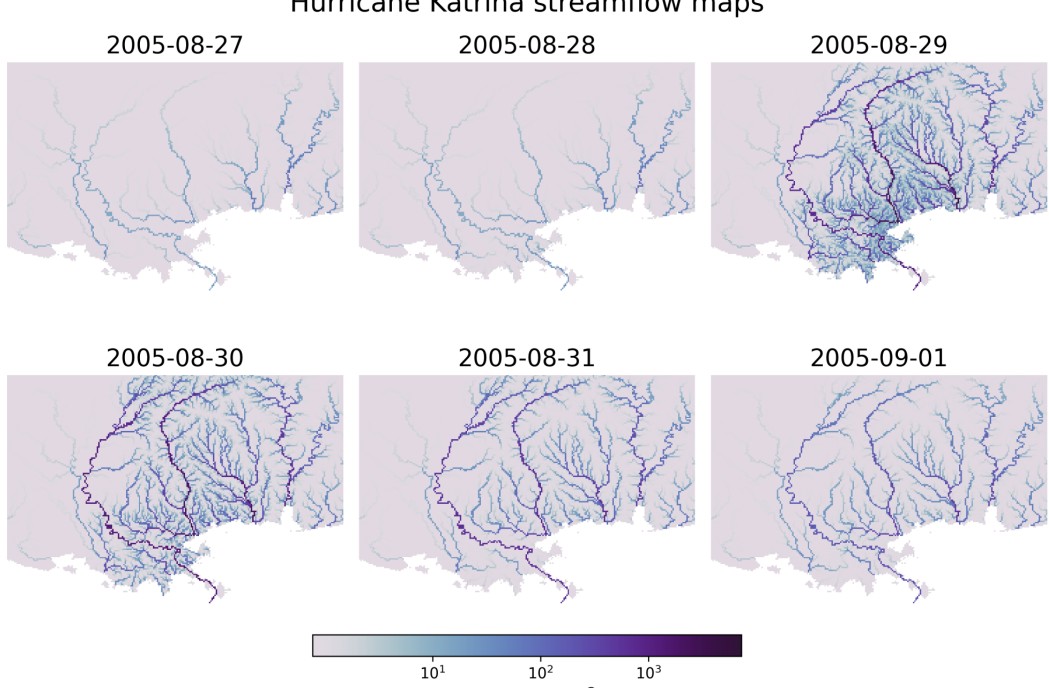

**Figure 12.** Streamflow dynamics during Hurricane Katrina from August 27 to September 1, 2005 for the ANN mapping. Each panel depicts the streamflow distribution across the affected region. To visualize the temporal evolution of the spatialized discharge pattern ; note that the kinematic wave routing was applied on flat topography, i.e. out of its validity range.

     ○ splitting the time window covered by hydrometric data into two complementary subsets over sub-periods of 4 years: $p1$ (from 1 August 2015 to 31 July 2019) and $p2$ (from 1 August 2019 to 31 July 2023), with $p1$ used as calibration period and $p2$ as validation period. For each period, 1 year is used as model "warm-up".

     ○ splitting the catchment set into 2 groups, upstream and downstream, calibrating on the upstream group and validating on the downstream group.

  – 3 calibration mappings across the whole Aude river are tested:

     ○ Uniform: spatially uniform parameters (gradient-free optimization)

     ○ Multi-Linear: a multiple linear regression is used as the transfer function from descriptors to spatialized parameters (gradient-based optimization)

     ○ ANN: a multi-layer perceptron composed of 3 hidden layers is used as the transfer function from descriptors to spatialized parameters (gradient-based optimization)

- The use of a multi-gauge cost function based on the average $NSE$ of the calibrated catchments ($J = \frac{1}{N_G} \sum^{N_G} (1 - NSE)$).

### 4.4.2 Results

The results of the regional mappings were validated on downstream gauges following Huynh et al. (2024b). The spatio-temporal validation performance, which assesses the model outside of the calibration gauges and period, is particularly challenging at such a high resolution and given the complex variabilities of physical factors and hydrological responses over this Mediterranean flash flood-prone case. The results are shown in validation only, for brevity again, in Figure 13. A uniform mapping yields a poor median $NSE$ of 0.15, while descriptors-to-parameters mappings achieve 0.62 and 0.69 for Multi-Linear and ANN approaches, respectively. Conceptual parameters maps obtained by learning from physical descriptors are shown in Figure 14 for the ANN mapping only (with the best $NSE$ result). The correlation matrix highlights significant correlations, especially between production capacity ($c_p$) and topographic slope ($d_5$), as well as exchange parameter ($k_{exc}$) or routing parameter ($a_{kw}$) with topographic slope ($d_5$). For each parameter, a correlation is also found with vegetation cover rate ($d_3$) or forest cover rate ($d_2$). This illustrates the interpretability of our neural network-based regionalization algorithm, in the space of conceptual model parameters. Regarding computation times, the calibration with ANN mapping over period $p1$ took 31 hours. This calibration involved 350 iterations, corresponding to 350 calls to the adjoint model, and was performed using 10 threads. For comparison, a single adjoint model run takes approximately 6 minutes, whereas a direct model run takes around 1 minute and 30 seconds using the same number of threads. A key feature of smash is its ability to accurately and efficiently compute spatially distributed cost gradients, as shown in Figure 16 in the conceptual parameter ($\boldsymbol{\theta}$) space for interpretability, in the case of a differentiable spatially distributed hydrological model including a NN-based regionalization mapping and a kinematic wave routing model (the partial differential equation numerical solver being also differentiated). Finally, simulated hydrographs are plotted for the six most downstream validation gauges in Figure 15 with a better reproduction for most downstream gauges in the present test configuration with the calibrated ANN regionalization (coherent with results of Huynh et al. (2024b) over the whole French Mediterranean region).

These performances are very encouraging since they were obtained with a relatively simple regionalization setup on a complex flash flood-prone area. Further research with smash will focus on improving the versatility of the hydrological model, to better account for high rainfall intensities (e.g. Daniela Peredo and Oudin (2022)) or groundwater/karstic effects, with classical or hybrid differential equations capable to learn from data at multiple scales, and to enrich the regionalization algorithms with advanced cost functions and spatial relaxation/regularization strategies. These improvements are necessary to better extract information, with the VDA algorithm, from multiple discharge gauges and other data sources (descriptors, satellite moisture, temperature, etc.). Additionally, incorporating more realistic hydraulic routing embedded within the differentiable hydrologic model will also enable the integration of hydraulic information (water levels, flow videos, etc), as introduced in Pujol et al. (2022).

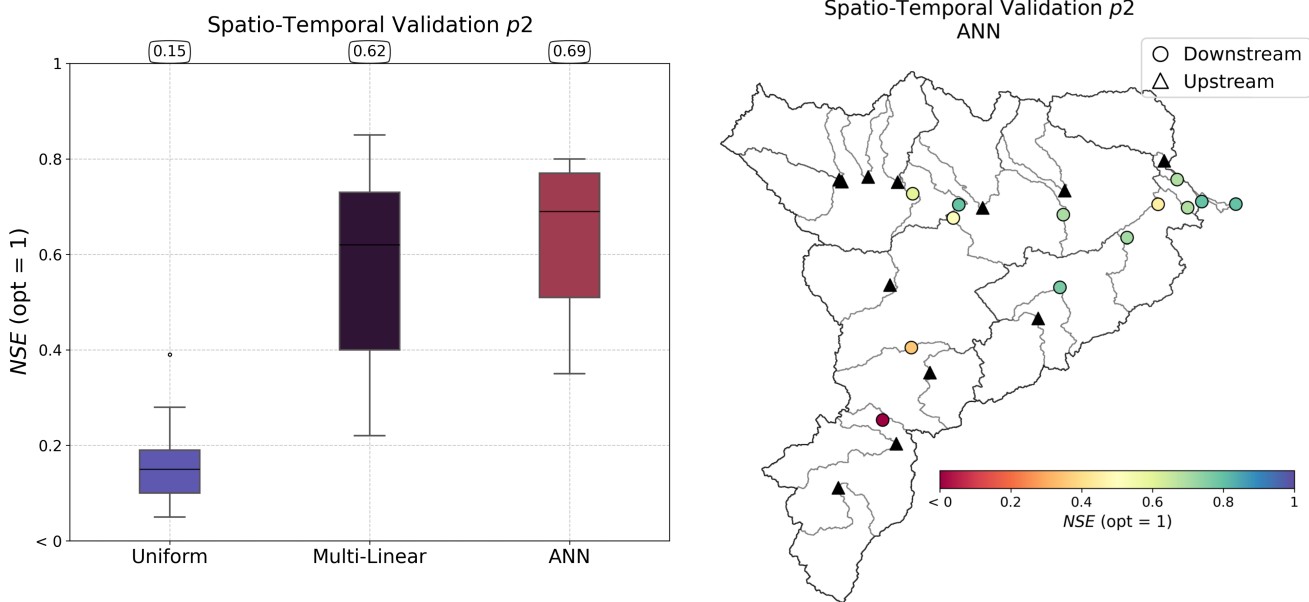

**Figure 13.** Performance in spatio-temporal validation over period $p2$ using calibration on upstream gauges (triangles). The boxplots on the left panel represent the distribution of Nash-Sutcliffe Efficiency ($NSE$) scores for the three calibration methods: Uniform, Multi-Linear, and Artificial Neural Network (ANN). Median values are displayed at the top of each boxplot. The map on the right illustrates the spatial distribution of the $NSE$ values for the ANN mapping for the downstream validation catchments.

## 5   Other `smash` features

In addition to the core differentiable spatialized hydrological solvers and regionalization learning algorithms illustrated above, `smash` enables performing:

- Automatic hydrograph segmentation and flood detection over large samples (Huynh et al., 2023).

- Parameter calibration using signature-based cost functions (Huynh et al., 2023) in addition to continuous metrics.

- Parameter calibration using a spatial regularization term (Jay-Allemand et al., 2020).

- Initial state estimation, including with regionalization mapping, even over short time windows which is applicable for short range VDA for operational forecasting

- Simulation of discharge ensembles from rainfall ensemble forecasting.

- Bayesian approach for parameter estimation and uncertainty quantification, with the consideration of structural model error and observation errors.

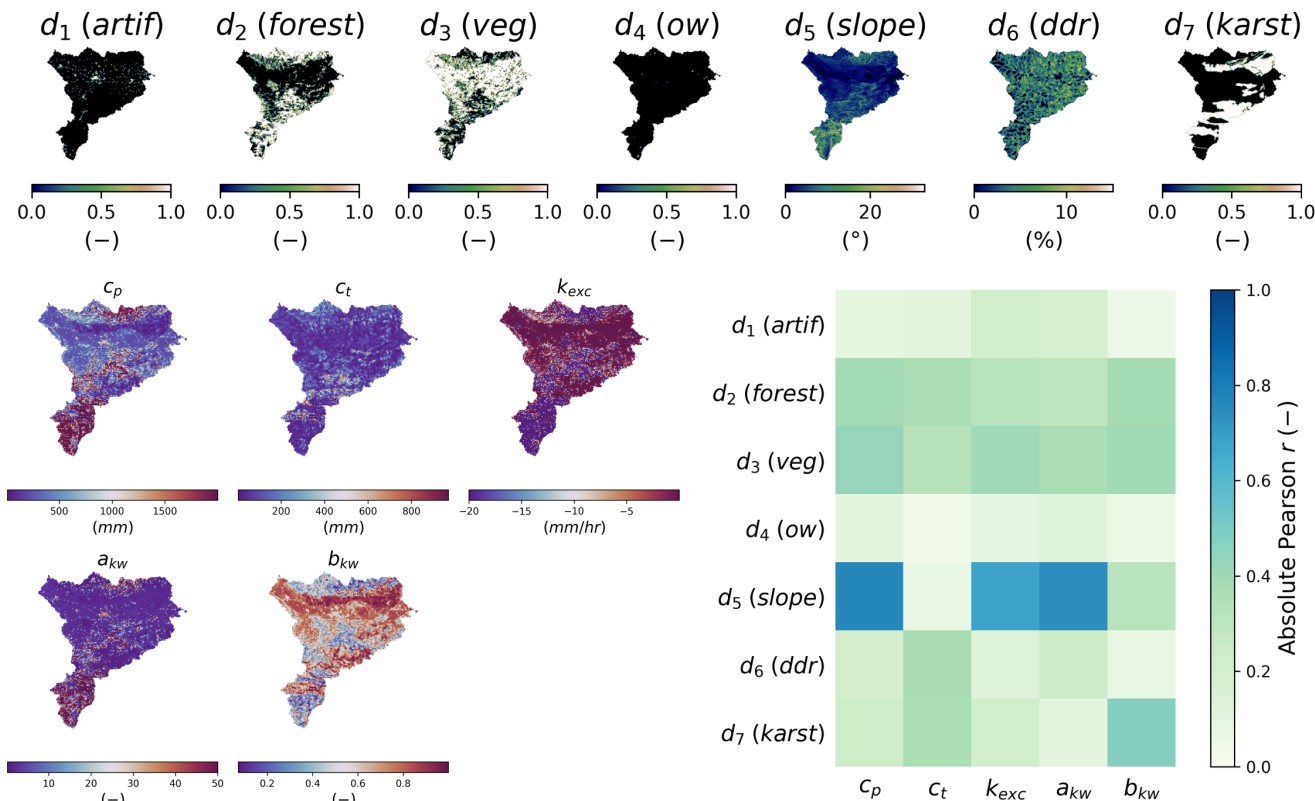

**Figure 14.** Analysis of input descriptors and output model parameters for the ANN descriptors-to-parameters mapping. Spatial distribution of physical descriptors ($d_1$-$d_7$) on the top panel, details provided in (Tab. 4); Spatial distribution of calibrated hydrological parameters ($c_p$, $c_t$, $k_{exc}$, $a_{kw}$, $b_{kw}$) on the lower left panel and linear correlation between descriptors and parameters on the lower right panel.

## 6  Conclusions

The recently released `smash` framework represents an advancement in modular, regionalizable, differentiable numerical modeling, as well as in hydrological data assimilation. This conclusion synthesizes the key principles, implementation features, performance indicators, and future prospects of `smash`, as presented in this article.

`smash` is built around three foundational principles: a modular operator chaining, enabling flexible representation of vertical and lateral hydrological processes; a regionalization mapping through hybrid approaches, combining conceptual models with descriptors-to-parameters neural networks; and a robust inverse algorithm that supports VDA.

The software leverages automatic differentiation to facilitate gradient-based calibration. Its seamless integration with Python via `f90wrap` ensures user-friendly access and flexibility, complemented by an automatic build system that simplifies deployment. Furthermore, `smash` supports parallel computing on CPUs, significantly accelerating computations for large-scale applications.

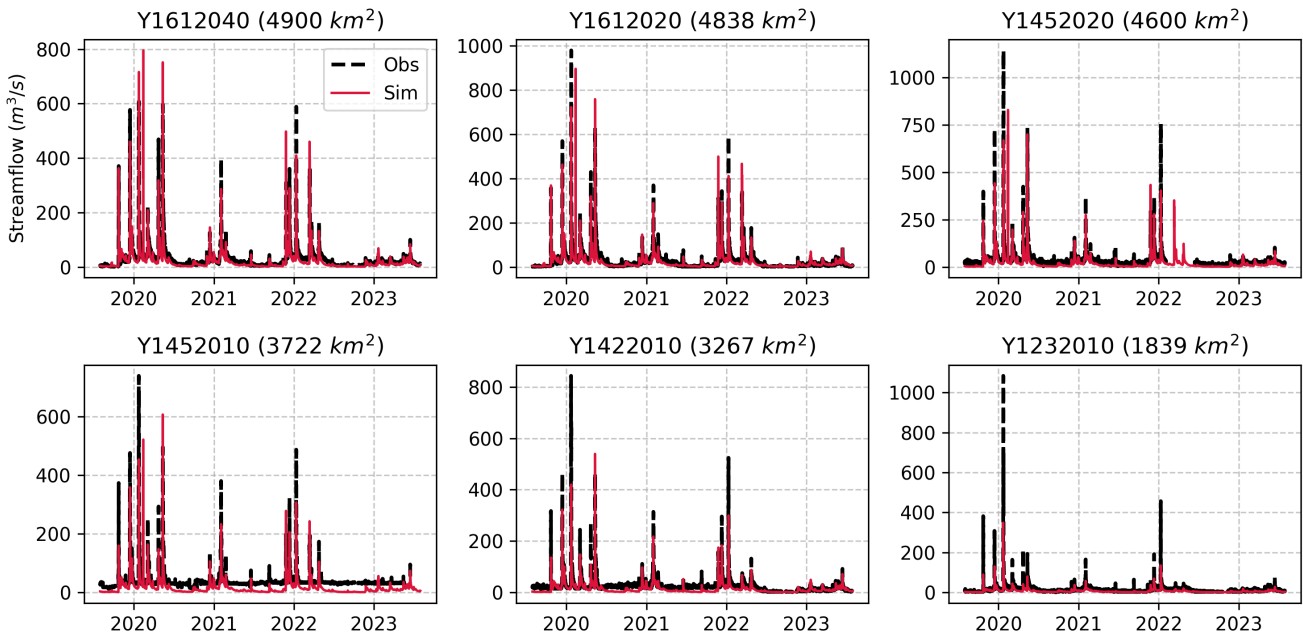

**Figure 15.** Observed and simulated streamflow of the six most downstream gauges of the Aude river for the ANN mapping. Each panel represents streamflow ($m^3/s$) for a specific gauge. Black dashed lines indicate observed values (Obs), while red solid lines represent simulated values (Sim).

In terms of hydrological modeling, `smash` achieves interesting results. Using CAMELS datasets, median $KGE > 0.8$ is observed in local spatially distributed calibration for daily GR-like and VIC-like model structures at $dx = 1'30''$ ($\sim 3km$). Additionally, regionalization learning across CONUS of conceptual parameters from physical descriptors yields $KGE > 0.6$ in spatio-temporal validation. High-resolution hourly modeling at $dx = 500m$ for Mediterranean flash-flood scenarios demonstrates $NSE > 0.6$.

`smash` planned enhancements include the integration of additional differentiable hydrological, hydraulic and land surface models, the expansion of hybrid physics-AI frameworks, and the refinement of data assimilation techniques. These advancements aim to further improve model accuracy, computational efficiency, and applicability in both research and operational settings.

*Code and data availability.* The source code of `smash`, Version 1.0, is available and preserved on multiple platforms: GitHub at https: //github.com/DassHydro/smash/tree/v1.0.2, PyPI at https://pypi.org/project/hydro-smash/1.0.2, and Zenodo with the DOI https://doi.org/10. 5281/zenodo.14841726 (Colleoni et al., 2025a). The datasets presented in this paper are also available on Zenodo under the DOI https:

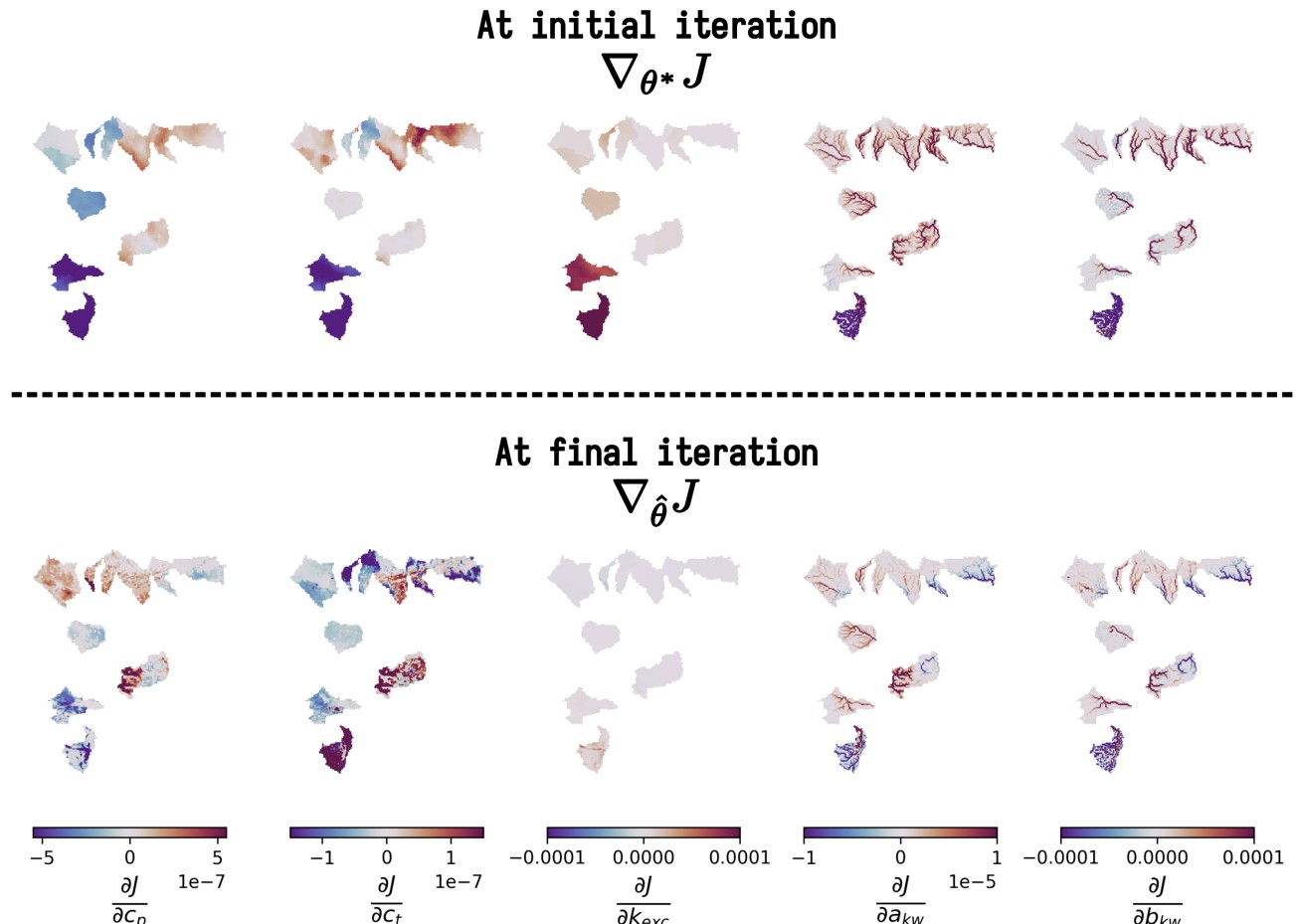

**Figure 16.** Spatially distributed gradients of the cost function $J$ with respect to the model parameters at initial and final iterations for ANN mapping. The first row shows the gradients $\nabla_{\theta^*} J$ at the initial iteration, while the second row presents the gradients $\nabla_{\hat{\theta}} J$ at the final iteration after optimization. Each column corresponds to the partial derivative of $J$ with respect to a specific parameter: $c_p$, $c_t$, $k_{exc}$, $a_{kw}$, and $b_{kw}$. These gradients are used in the optimization process of the control vector $\rho$ using $\nabla_{\rho} J = \nabla_{\theta} J . \nabla_{\rho} \theta$ with $\theta = \mathcal{N}(., \rho)$ where $\mathcal{N}$ is the multi-layer perceptron used.

//doi.org/10.5281/zenodo.14865491 (Colleoni et al., 2025b). `smash` is released under the GPL-3 license and is developed openly at https://github.com/DassHydro/smash. The documentation is accessible at https://smash.recover.inrae.fr.

 **Appendix A:  CONUS - CAMELS - Split-sample temporal cross-validation**

**Table A1.** Summary of model operators and their associated parameters. For each operator, the parameter name, description, initial value, and allowable value range are provided.

| Operator name | Parameter name | Parameter description | Parameter initial value | Parameter range |
|---|---|---|---|---|
| $\mathcal{M}_{snw}$: *ssn* | $k_{mlt}$ | Melt coefficient $(mm \cdot {}^{\circ}C^{-1} \cdot day^{-1})$ | 1 | [0.01, 100] |
| $\mathcal{M}_{rr}$: *gr4* | $c_p$ | Maximum capacity of the production reservoir $(mm)$ | 200 | [1, 2000] |
| | $c_t$ | Maximum capacity of the transfer reservoir $(mm)$ | 500 | [1, 2000] |
| | $k_{exc}$ | Exchange coefficient $(mm \cdot day^{-1})$ | 0 | [-50, 50] |
| $\mathcal{M}_{rr}$: *gr5* | $c_p$ | Maximum capacity of the production reservoir $(mm)$ | 200 | [1, 2000] |
| | $c_t$ | Maximum capacity of the transfer reservoir $(mm)$ | 500 | [1, 2000] |
| | $k_{exc}$ | Exchange coefficient $(mm \cdot day^{-1})$ | 0 | [-50, 50] |
| | $a_{exc}$ | Exchange threshold $(-)$ | 0 | [0.001, 0.999] |
| $\mathcal{M}_{rr}$: *grd* | $c_p$ | Maximum capacity of the production reservoir $(mm)$ | 200 | [1, 5000] |
| | $c_t$ | Maximum capacity of the transfer reservoir $(mm)$ | 500 | [1, 5000] |
| $\mathcal{M}_{rr}$: *loieau* | $c_a$ | Maximum capacity of the production reservoir $(mm)$ | 200 | [1, 2000] |
| | $c_c$ | Maximum capacity of the transfer reservoir $(mm)$ | 500 | [1, 2000] |
| | $k_b$ | Transfer coefficient $(-)$ | 1 | [0.01, 4] |
| $\mathcal{M}_{rr}$: *vic3l* | $b$ | Variable infiltration curve parameter $(-)$ | 0.1 | [0.001, 0.4] |
| | $c_{usl}$ | Maximum capacity of the upper soil layer $(mm)$ | 100 | [10, 500] |
| | $c_{msl}$ | Maximum capacity of the medium soil layer $(mm)$ | 100 | [50, 1000] |
| | $c_{bsl}$ | Maximum capacity of the bottom soil layer $(mm)$ | 100 | [500, 2500] |
| | $k_s$ | Saturated hydraulic conductivity $(mm \cdot day^{-1})$ | 20 | Not optimized |
| | $p_{bc}$ | Brooks and Corey exponent $(-)$ | 10 | Not optimized |
| | $d_s$ | Non-linear baseflow threshold maximum velocity $(-)$ | 0.01 | [0.001, 1] |
| | $d_{sm}$ | Maximum velocity of baseflow $(mm \cdot day^{-1})$ | 0.5 | [0.2, 1] |
| | $w_s$ | Non-linear baseflow threshold soil moisture $(-)$ | 0.8 | [0.1, 1] |
| $\mathcal{M}_{hy}$: *kw* | $a_{kw}$ | Alpha kinematic wave parameter $(-)$ | 5 | [0.001, 50] |
| | $b_{kw}$ | Beta kinematic wave parameter $(-)$ | 0.6 | [0.001, 1] |

**Table A2.** Summary statistics of the calibrated parameters across the set of 482 catchments. For each parameter, the median, standard deviation, and coefficient of variation ($\mu/\sigma$) are reported for the two calibration configurations : spatially uniform ("Uniform") and spatially distributed ("Distributed"). For the spatially distributed calibration, statistics were computed based on the spatial average of parameter values within each catchment.

| Model name | Parameter name | Median (Uniform ∥ Distributed) | Standard deviation (Uniform ∥ Distributed) | Coefficient of variation (Uniform ∥ Distributed) |
|---|---|---|---|---|
| gr4 | $k_{mlt}$ | 1.49 ∥ 1.68 | 18.30 ∥ 18.34 | 3.14 ∥ 2.92 |
| | $c_p$ | 96.27 ∥ 104.98 | 247.65 ∥ 240.72 | 1.39 ∥ 1.37 |
| | $c_t$ | 24.86 ∥ 30.42 | 278.39 ∥ 262.29 | 2.56 ∥ 2.41 |
| | $k_{exc}$ | 0.19 ∥ 0.05 | 7.70 ∥ 7.96 | -6.77 ∥ -5.01 |
| | $a_{kw}$ | 3.61 ∥ 3.55 | 13.57 ∥ 13.57 | 1.51 ∥ 1.52 |
| | $b_{kw}$ | 0.34 ∥ 0.34 | 0.42 ∥ 0.42 | 0.96 ∥ 0.96 |
| gr5 | $k_{mlt}$ | 1.43 ∥ 1.64 | 17.72 ∥ 17.67 | 3.25 ∥ 3.09 |
| | $c_p$ | 90.03 ∥ 89.75 | 211.25 ∥ 201.84 | 1.32 ∥ 1.32 |
| | $c_t$ | 23.01 ∥ 31.56 | 317.23 ∥ 310.26 | 2.72 ∥ 2.60 |
| | $k_{exc}$ | 0.06 ∥ -0.01 | 4.52 ∥ 4.90 | -21.66 ∥ -15.81 |
| | $a_{exc}$ | 0.14 ∥ 0.13 | 0.20 ∥ 0.20 | 0.96 ∥ 0.97 |
| | $a_{kw}$ | 3.97 ∥ 3.81 | 15.64 ∥ 15.64 | 1.45 ∥ 1.46 |
| | $b_{kw}$ | 0.34 ∥ 0.34 | 0.41 ∥ 0.41 | 0.95 ∥ 0.95 |
| grd | $k_{mlt}$ | 1.39 ∥ 1.76 | 18.24 ∥ 18.32 | 3.23 ∥ 2.91 |
| | $c_p$ | 62.13 ∥ 95.86 | 617.23 ∥ 715.60 | 2.53 ∥ 2.17 |
| | $c_t$ | 38.84 ∥ 44.99 | 356.07 ∥ 384.23 | 2.84 ∥ 2.59 |
| | $a_{kw}$ | 1.89 ∥ 1.96 | 18.74 ∥ 18.72 | 1.55 ∥ 1.55 |
| | $b_{kw}$ | 0.34 ∥ 0.33 | 0.42 ∥ 0.42 | 0.92 ∥ 0.92 |
| loieau | $k_{mlt}$ | 1.46 ∥ 1.65 | 18.10 ∥ 18.14 | 3.13 ∥ 2.94 |
| | $c_a$ | 134.24 ∥ 306.95 | 303.37 ∥ 320.17 | 1.33 ∥ 0.82 |
| | $c_c$ | 14.27 ∥ 72.51 | 333.68 ∥ 375.33 | 2.81 ∥ 1.67 |
| | $k_b$ | 1.10 ∥ 1.17 | 0.55 ∥ 0.55 | 0.48 ∥ 0.47 |
| | $a_{kw}$ | 2.66 ∥ 3.65 | 14.36 ∥ 14.35 | 1.64 ∥ 1.48 |
| | $b_{kw}$ | 0.38 ∥ 0.38 | 0.43 ∥ 0.41 | 0.92 ∥ 0.89 |
| vic3l | $k_{mlt}$ | 1.62 ∥ 1.98 | 18.23 ∥ 18.15 | 2.86 ∥ 2.59 |
| | $b$ | 0.15 ∥ 0.14 | 0.15 ∥ 0.15 | 0.80 ∥ 0.83 |
| | $c_{usl}$ | 84.36 ∥ 86.63 | 122.29 ∥ 121.54 | 1.03 ∥ 1.00 |
| | $c_{msl}$ | 170.56 ∥ 178.35 | 361.36 ∥ 359.62 | 1.01 ∥ 1.01 |
| | $c_{bsl}$ | 1799.74 ∥ 1799.74 | 704.74 ∥ 702.97 | 0.44 ∥ 0.44 |
| | $d_s$ | 0.12 ∥ 0.12 | 0.33 ∥ 0.32 | 1.26 ∥ 1.24 |
| | $d_{sm}$ | 0.50 ∥ 0.50 | 0.23 ∥ 0.23 | 0.45 ∥ 0.45 |
| | $w_s$ | 0.80 ∥ 0.80 | 0.26 ∥ 0.26 | 0.34 ∥ 0.34 |
| | $a_{kw}$ | 50.00 ∥ 49.45 | 19.93 ∥ 19.89 | 0.60 ∥ 0.60 |
| | $b_{kw}$ | 0.44 ∥ 0.43 | 0.31 ∥ 0.30 | 0.64 ∥ 0.64 |

## Appendix B: CONUS - CAMELS - Regionalization

**Table B1.** Summary of model operators and their associated parameters. For each operator, the parameter name, description, initial value, and allowable value range are provided.

| Operator name | Parameter name | Parameter description | Parameter initial value | Parameter range |
|---|---|---|---|---|
| $\mathcal{M}_{snw}$: *ssn* | $k_{mlt}$ | Melt coefficient ($mm \cdot {}^{\circ}C^{-1} \cdot day^{-1}$) | 1 | [0.01, 100] |
| $\mathcal{M}_{rr}$: *gr4* | $c_p$ | Maximum capacity of the production reservoir ($mm$) | 200 | [1, 2000] |
| | $c_t$ | Maximum capacity of the transfer reservoir ($mm$) | 500 | [1, 2000] |
| | $k_{exc}$ | Exchange coefficient ($mm \cdot day^{-1}$) | 0 | [-50, 0] |
| $\mathcal{M}_{hy}$: *kw* | $a_{kw}$ | Alpha kinematic wave parameter ($-$) | 5 | [0.001, 50] |
| | $b_{kw}$ | Beta kinematic wave parameter ($-$) | 0.6 | [0.001, 1] |

**Table B2.** Median $KGE$ obtained in regionalization mapping calibration-validation over 4 groups of randomly selected basins.

| Group | Calibration $KGE_{50}$ | Spatial Validation $KGE_{50}$ | Temporal Validation $KGE_{50}$ | Spatio-Temporal Validation $KGE_{50}$ |
|---|---|---|---|---|
| 0 | 0.65 | 0.62 | 0.65 | 0.65 |
| 1 | 0.62 | 0.58 | 0.65 | 0.58 |
| 2 | 0.65 | 0.65 | 0.64 | 0.67 |
| 3 | 0.65 | 0.63 | 0.65 | 0.63 |

## Appendix C: France - Aude - High-resolution Regionalization

**Table C1.** Summary of model operators and their associated parameters. For each operator, the parameter name, description, initial value, and allowable value range are provided.

| Operator name | Parameter name | Parameter description | Parameter initial value | Parameter range |
|---|---|---|---|---|
| $\mathcal{M}_{rr}$: *gr4* | $c_p$ | Maximum capacity of the production reservoir ($mm$) | 200 | [1, 2000] |
| | $c_t$ | Maximum capacity of the transfer reservoir ($mm$) | 500 | [1, 2000] |
| | $k_{exc}$ | Exchange coefficient ($mm \cdot hour^{-1}$) | 0 | [-20, 0] |
| $\mathcal{M}_{hy}$: *kw* | $a_{kw}$ | Alpha kinematic wave parameter ($-$) | 5 | [0.001, 50] |
| | $b_{kw}$ | Beta kinematic wave parameter ($-$) | 0.6 | [0.001, 1] |

## Appendix D: `smash` operators

This section describes the various operators available in `smash` with mathematical/numerical expressions, **input data** $[\boldsymbol{I}, \boldsymbol{D}](x,t)$, **tunable conceptual parameters** $\boldsymbol{\theta}(x,t)$, and simulated **states and fluxes** $\boldsymbol{U}(x,t) = [Q, \boldsymbol{h}, \boldsymbol{q}](x,t)$. These operators are written below for a given pixel $x$ of the 2D spatial domain $\Omega$ and for a time $t$ in the simulation window $]0, T]$.

### D1   Snow operator $\mathcal{M}_{snw}$

#### ○ *zero*

This snow operator simply means that there is no snow operator.

$$m_{lt}(x,t) = 0$$

with $m_{lt}$ the melt flux.

#### ○ *ssn*

This snow operator is a simple degree-day snow operator.

Update the snow reservoir state $h_s$ for $t^* \in ]t-1, t[$

$$h_s(x, t^*) = h_s(x, t-1) + S(x,t)$$

Compute the melt flux $m_{lt}$

$$m_{lt}(x,t) = \begin{cases} 0 & \text{if } T_e(x,t) \leq 0 \\ \min\left(h_s(x, t^*), k_{mlt}(x) \cdot T_e(x,t)\right) & \text{otherwise.} \end{cases}$$

Update the snow reservoir state $h_s$

$$h_s(x,t) = h_s(x, t^*) - m_{lt}(x,t)$$

with $m_{lt}$ the melt flux, $S$ the snow, $T_e$ the temperature, $k_{mlt}$ the melt coefficient and $h_s$ the state of the snow reservoir.

### D2   Hydrological operator $\mathcal{M}_{hy}$

#### ○ *gr4*

This hydrological operator is derived from the GR4 model (Perrin et al., 2003)

**Interception**

Compute interception evapotranspiration $e_i$:

$$e_i(x,t) = \min\left(E(x,t),\ P(x,t) + m_{lt}(x,t) + \tilde{h}_i(x, t-1) \cdot c_i(x)\right)$$

Compute the neutralized precipitation $p_n$ and evapotranspiration $e_n$:

$$p_n(x,t) = \max\left(0,\ P(x,t) + m_{lt}(x,t) - c_i(x)(1 - \tilde{h}_i(x,t-1)) - e_i(x,t)\right)$$

$$e_n(x,t) = E(x,t) - e_i(x,t)$$

Update the normalized interception reservoir state $\tilde{h}_i$:

$$\tilde{h}_i(x,t) = \tilde{h}_i(x,t-1) + \frac{P(x,t) + m_{lt}(x,t) + e_i(x,t) - p_n(x,t)}{c_i(x)}$$

### Production

Compute the production infiltrating precipitation $p_s$ and evapotranspiration $e_s$:

$$p_s(x,t) = c_p(x)\left(1 - \tilde{h}_p(x,t-1)^2\right) \cdot \frac{\tanh\left(\frac{p_n(x,t)}{c_p(x)}\right)}{1 + \tilde{h}_p(x,t-1)\tanh\left(\frac{p_n(x,t)}{c_p(x)}\right)}$$

$$e_s(x,t) = \tilde{h}_p(x,t-1) \cdot c_p(x) \cdot \left(2 - \tilde{h}_p(x,t-1)\right) \cdot \frac{\tanh\left(\frac{e_n(x,t)}{c_p(x)}\right)}{1 + \left(1 - \tilde{h}_p(x,t-1)\right)\tanh\left(\frac{e_n(x,t)}{c_p(x)}\right)}$$

Update the normalized production reservoir state $\tilde{h}_p$:

$$\tilde{h}_p(x,t^*) = \tilde{h}_p(x,t-1) + \frac{p_s(x,t) - e_s(x,t)}{c_p(x)}$$

Compute the production runoff $p_r$:

$$p_r(x,t) = \begin{cases} 0 & \text{if } p_n(x,t) \leq 0 \\ p_n(x,t) - \left(\tilde{h}_p(x,t^*) - \tilde{h}_p(x,t-1)\right)c_p(x) & \text{otherwise} \end{cases}$$

Compute the production percolation $p_{erc}$:

$$p_{erc}(x,t) = \tilde{h}_p(x,t^*) \cdot c_p(x)\left(1 - \left(1 + \left(\frac{4}{9}\tilde{h}_p(x,t^*)\right)^4\right)^{-1/4}\right)$$

Update the normalized production reservoir state $\tilde{h}_p$:

$$\tilde{h}_p(x,t) = \tilde{h}_p(x,t^*) - \frac{p_{erc}(x,t)}{c_p(x)}$$

515 **Exchange**

Compute the exchange flux $l_{exc}$:

$$l_{exc}(x,t) = k_{exc}(x) \cdot \tilde{h}_t(x,t-1)^{7/2}$$

**Transfer**

Split the production runoff $p_r$ into two branches (transfer and direct), $p_{rr}$ and $p_{rd}$:

520 $$p_{rr}(x,t) = 0.9\,(p_r(x,t) + p_{erc}(x,t)) + l_{exc}(x,t)$$

$$p_{rd}(x,t) = 0.1\,(p_r(x,t) + p_{erc}(x,t))$$

Update the normalized transfer reservoir state $\tilde{h}_t$:

$$\tilde{h}_t(x,t^*) = \max\left(0,\ \tilde{h}_t(x,t-1) + \frac{p_{rr}(x,t)}{c_t(x)}\right)$$

525

Compute the transfer branch elemental discharge $q_r$:

$$q_r(x,t) = \tilde{h}_t(x,t^*) \cdot c_t(x) - \left(\left(\tilde{h}_t(x,t^*) \cdot c_t(x)\right)^{-4} + c_t(x)^{-4}\right)^{-1/4}$$

Update the normalized transfer reservoir state $\tilde{h}_t$:

530 $$\tilde{h}_t(x,t) = \tilde{h}_t(x,t^*) - \frac{q_r(x,t)}{c_t(x)}$$

Compute the direct branch elemental discharge $q_d$:

$$q_d(x,t) = \max\left(0,\ p_{rd}(x,t) + l_{exc}(x,t)\right)$$

535 Compute the elemental discharge $q_t$:

$$q_t(x,t) = q_r(x,t) + q_d(x,t)$$

with $q_t$ the elemental discharge, $P$ the precipitation, $E$ the potential evapotranspiration, $m_{lt}$ the melt flux from the snow operator, $c_i$ the maximum capacity of the interception reservoir, $c_p$ the maximum capacity of the production reservoir, $c_t$ the maximum capacity of the transfer reservoir, $k_{exc}$ the exchange coefficient, $\tilde{h}_i$ the state of the normalized interception reservoir, 540 $\tilde{h}_p$ the state of the normalized production reservoir and $\tilde{h}_t$ the state of the normalized transfer reservoir.

○ **gr5**

This hydrological operator is derived from the GR4 model (Le Moine, 2008). It consists in a GR4 like model structure (see above) with a modified exchange flux with two parameters to account for seasonal variations.

**545** **Interception**

Same as *gr4* Interception

**Production**

Same as *gr4* Production

**Exchange**

**550** Compute the exchange flux $l_{exc}$:

$$l_{exc}(x,t) = k_{exc}(x) \cdot \tilde{h}_t(x,t-1)^{7/2}$$

**Transfer**

Same as *gr4* Transfer

with $q_t$ the elemental discharge, $P$ the precipitation, $E$ the potential evapotranspiration, $m_{lt}$ the melt flux from the snow op-
**555** erator, $c_i$ the maximum capacity of the interception reservoir, $c_p$ the maximum capacity of the production reservoir, $c_t$ the maximum capacity of the transfer reservoir, $k_{exc}$ the exchange coefficient, $a_{exc}$ the exchange threshold, $\tilde{h}_i$ the state of the normalized interception reservoir, $\tilde{h}_p$ the state of the normalized production reservoir and $\tilde{h}_t$ the state of the normalized transfer reservoir.

**560** ○ **grd**

This hydrological operator is derived from the GR models and is a simplified structure used in Jay-Allemand et al. (2020).

**Interception**

Compute the interception evapotranspiration $e_i$:

$$e_i(x,t) = \min\left(E(x,t),\ P(x,t) + m_{lt}(x,t)\right)$$

**565**

Compute the neutralized precipitation $p_n$ and evapotranspiration $e_n$:

$$p_n(x,t) = \max\left(0,\ P(x,t) + m_{lt}(x,t) - e_i(x,t)\right)$$
$$e_n(x,t) = E(x,t) - e_i(x,t)$$

**570** **Production**

Same as *gr4* Production

**Transfer**

Update the normalized transfer reservoir state $\tilde{h}_t$:

$$\tilde{h}_t(x,t^*) = \max\left(0, \ \tilde{h}_t(x,t-1) + \frac{p_r(x,t)}{c_t(x)}\right)$$


Compute the transfer branch elemental discharge $q_r$:

$$q_r(x,t) = \tilde{h}_t(x,t^*) \cdot c_t(x) - \left(\left(\tilde{h}_t(x,t^*) \cdot c_t(x)\right)^{-4} + c_t(x)^{-4}\right)^{-1/4}$$

Update the normalized transfer reservoir state $\tilde{h}_t$:

$\quad \tilde{h}_t(x,t) = \tilde{h}_t(x,t^*) - \dfrac{q_r(x,t)}{c_t(x)}$

Compute the elemental discharge $q_t$:

$$q_t(x,t) = q_r(x,t)$$

with $q_t$ the elemental discharge, $P$ the precipitation, $E$ the potential evapotranspiration, $m_{lt}$ the melt flux from the snow oper-
ator, $c_p$ the maximum capacity of the production reservoir, $c_t$ the maximum capacity of the transfer reservoir, $\tilde{h}_p$ the state of
the normalized production reservoir and $\tilde{h}_t$ the state of the normalized transfer reservoir.

○ *loieau*

This hydrological operator is derived from the GR model (Folton and Arnaud, 2020).

**Interception**

Same as *gr4* Interception

**Production**

Same as *gr4* Production

**Transfer**

Split the production runoff $p_r$ into two branches (transfer and direct), $p_{rr}$ and $p_{rd}$:

$$p_{rr}(x,t) = 0.9\left(p_r(x,t) + p_{erc}(x,t)\right)$$
$$p_{rd}(x,t) = 0.1\left(p_r(x,t) + p_{erc}(x,t)\right)$$

Update the normalized transfer reservoir state $\tilde{h}_c$:

$\quad \tilde{h}_c(x,t^*) = \max\left(0, \ \tilde{h}_c(x,t-1) + \dfrac{p_{rr}(x,t)}{c_c(x)}\right)$

Compute the transfer branch elemental discharge $q_r$:

$$q_r(x,t) = \tilde{h}_c(x,t^*) \cdot c_c(x) - \left( \left( \tilde{h}_c(x,t^*) \cdot c_c(x) \right)^{-3} + c_c(x)^{-3} \right)^{-1/3}$$

Update the normalized transfer reservoir state $\tilde{h}_c$:

$$\tilde{h}_c(x,t) = \tilde{h}_c(x,t^*) - \frac{q_r(x,t)}{c_c(x)}$$

Compute the direct branch elemental discharge $q_d$:

$$q_d(x,t) = \max\left(0,\ p_{rd}(x,t)\right)$$


Compute the elemental discharge $q_t$:

$$q_t(x,t) = k_b(x) \cdot (q_r(x,t) + q_d(x,t))$$

with $q_t$ the elemental discharge, $P$ the precipitation, $E$ the potential evapotranspiration, $m_{lt}$ the melt flux from the snow oper-
ator, $c_a$ the maximum capacity of the production reservoir, $c_c$ the maximum capacity of the transfer reservoir, $k_b$ the transfer
coefficient, $\tilde{h}_a$ the state of the normalized production reservoir and $\tilde{h}_c$ the state of the normalized transfer reservoir.

   ○ *vic3l*
This hydrological operator is derived from the VIC model (Liang et al., 1994).
**Canopy Layer Interception**
Compute the canopy layer interception evapotranspiration $e_c$:

$$e_c(x,t) = \min\left( E(x,t)\tilde{h}_{cl}(x,t-1)^{2/3},\ P(x,t) + m_{lt}(x,t) + \tilde{h}_{cl}(x,t-1) \right)$$

Compute the neutralized precipitation $p_n$ and evapotranspiration $e_n$:

$$p_n(x,t) = \max\left( 0,\ P(x,t) + m_{lt}(x,t) - (1 - \tilde{h}_{cl}(x,t-1)) - e_c(x,t) \right)$$

$e_n(x,t) = E(x,t) - e_c(x,t)$

Update the normalized canopy layer interception state $\tilde{h}_{cl}$:

$$\tilde{h}_{cl}(x,t) = \tilde{h}_{cl}(x,t-1) + P(x,t) - e_c(x,t) - p_n(x,t)$$

**Upper Soil Layer Evapotranspiration**

Compute the maximum infiltration $i_m$ and the corresponding soil saturation infiltration $i_0$:

$$i_m(x,t) = (1 + b(x))c_{usl}(x)$$

$$i_0(x,t) = i_m(x,t)\left(1 - (1 - \tilde{h}_{usl}(x,t-1))^{1/(1-b(x))}\right)$$

Compute the upper soil layer evapotranspiration $e_s$:

$$e_s(x,t) = \begin{cases} e_n(x,t) & \text{if } i_0(x,t) \geq i_m(x,t) \\ \beta(x,t)e_n(x,t) & \text{otherwise} \end{cases}$$

with $\beta$, the ARNO evapotranspiration beta function (Todini, 1996).

Update the normalized upper soil layer reservoir state $\tilde{h}_{usl}$:

$$\tilde{h}_{usl}(x,t) = \tilde{h}_{usl}(x,t-1) - \frac{e_s(x,t)}{c_{usl}(x)}$$

**Infiltration**

Compute the maximum capacity $c_{umsl}$, soil moisture $w_{umsl}$, and relative state $h_{umsl}$ of the first two layers:

$$c_{umsl}(x) = c_{usl}(x) + c_{msl}(x)$$

$$w_{umsl}(x,t-1) = \tilde{h}_{usl}(x,t-1)c_{usl}(x) + \tilde{h}_{msl}(x,t-1)c_{msl}(x)$$

$$h_{umsl}(x,t-1) = \frac{w_{umsl}(x,t-1)}{c_{umsl}(x)}$$

Compute maximum $i_m$ and infiltration $i_0$:

$$i_m(x,t) = (1 + b(x))c_{umsl}(x)$$

$$i_0(x,t) = i_m(x,t)\left(1 - (1 - h_{umsl}(x,t-1))^{1/(1-b(x))}\right)$$

Compute infiltration $i$:

$$i(x,t) = \begin{cases} c_{umsl}(x) - w_{umsl}(x,t-1) & \text{if } i_0(x,t) + p_n(x,t) > i_m(x,t) \\ c_{umsl}(x) - w_{umsl}(x,t-1) - c_{umsl}(x)\left(1 - \frac{i_0(x,t)+p_n(x,t)}{i_m(x,t)}\right)^{b(x)+1} & \text{otherwise} \end{cases}$$

Distribute infiltration between the two upper layers:

$$i_{usl}(x,t) = \min\left((1 - \tilde{h_{usl}}(x,t-1))c_{usl}(x),\ i(x,t)\right)$$

$$i_{msl}(x,t) = \min\left((1 - \tilde{h_{msl}}(x,t-1))c_{msl}(x),\ i(x,t) - i_{usl}(x,t)\right)$$

Update the reservoir states:

$$\tilde{h_{usl}}(x,t) = \tilde{h_{usl}}(x,t-1) + i_{usl}(x,t)$$

$$\tilde{h_{msl}}(x,t) = \tilde{h_{msl}}(x,t-1) + i_{msl}(x,t)$$

Compute runoff:

$$q_r(x,t) = p_n(x,t) - (i_{usl}(x,t) + i_{msl}(x,t))$$

**Drainage**

Compute the soil moisture in the first two layers:

$$w_{usl}(x,t-1) = \tilde{h_{usl}}(x,t-1)c_{usl}(x)$$

$$w_{msl}(x,t-1) = \tilde{h_{msl}}(x,t-1)c_{msl}(x)$$

Compute the initial drainage flux:

$$d_{umsl}(x,t^*) = k_s(x) \cdot \tilde{h_{usl}}(x,t-1)^{p_{bc}}$$

Update the drainage flux:

$$d_{umsl}(x,t) = \min\left(d_{umsl}(x,t^*),\ \min(w_{usl}(x,t-1),\ c_{msl}(x) - w_{msl}(x,t-1))\right)$$

Update normalized reservoir states:

$$\tilde{h_{usl}}(x,t) = \tilde{h_{usl}}(x,t-1) - \frac{d_{umsl}(x,t)}{c_{usl}(x)}$$

$$\tilde{h_{msl}}(x,t) = \tilde{h_{msl}}(x,t-1) + \frac{d_{umsl}(x,t)}{c_{msl}(x)}$$

The same approach is performed for drainage between medium and bottom soil layers. For brevity, we skip the first steps and directly give the update equations.

Update of the normalized medium and bottom reservoir states:

$$\tilde{h_{msl}}(x,t) = \tilde{h_{msl}}(x,t-1) - \frac{d_{mbsl}(x,t)}{c_{msl}(x)}$$

$$\tilde{h_{bsl}}(x,t) = \tilde{h_{bsl}}(x,t-1) + \frac{d_{mbsl}(x,t)}{c_{bsl}(x)}$$

**Baseflow**

Compute baseflow $q_b$:

$$q_b(x,t) = \begin{cases} \frac{d_{sm}(x)d_s(x)}{w_s(x)} \tilde{h_{bsl}}(x,t-1) & \text{if } \tilde{h_{bsl}}(x,t-1) \leq w_s(x) \\ \frac{d_{sm}(x)d_s(x)}{w_s(x)} \tilde{h_{bsl}}(x,t-1) + d_{sm}(x) \left(1 - \frac{d_s(x)}{w_s(x)}\right) \left(\frac{\tilde{h_{bsl}}(x,t-1) - w_s(x)}{1 - w_s(x)}\right)^2 & \text{otherwise} \end{cases}$$

Update the normalized bottom soil layer reservoir:

$$\tilde{h_{bsl}}(x,t) = \tilde{h_{bsl}}(x,t-1) - \frac{q_b(x,t)}{c_{bsl}(x)}$$

with $q_t$ the elemental discharge, $P$ the precipitation, $E$ the potential evapotranspiration, $m_{lt}$ the melt flux from the snow operator, $b$ the variable infiltration curve parameter, $c_{usl}$ the maximum capacity of the upper soil layer, $c_{msl}$ the maximum capacity of the medium soil layer, $c_{bsl}$ the maximum capacity of the bottom soil layer, $k_s$ the saturated hydraulic conductivity, $p_{bc}$ the Brooks and Corey exponent, $d_{sm}$ the maximum velocity of baseflow, $d_s$ the non-linear baseflow threshold maximum velocity, $w_s$ the non-linear baseflow threshold soil moisture, $\tilde{h_{cl}}$ the state of the normalized canopy layer, $\tilde{h_{usl}}$ the state of the normalized upper soil layer, $\tilde{h_{msl}}$ the state of the normalized medium soil layer and $\tilde{h_{bsl}}$ the state of the normalized bottom soil layer.

## D3 Routing operator $\mathcal{M}_{rr}$

○ *lag0*

This routing operator is a simple aggregation of upstream discharge to downstream following the drainage plan.

**Upstream Discharge**

Compute the upstream discharge $q_{up}$:

$$q_{up}(x,t) = \begin{cases} 0 & \text{if } \Omega_x = \emptyset \\ \sum_{k \in \Omega_x} Q(k,t) & \text{otherwise} \end{cases}$$

where $\Omega_x$ is the set of upstream cells flowing into cell $x$.


**Surface Discharge**

Compute the surface discharge $Q$:

$$Q(x,t) = q_{up}(x,t) + \alpha(x)\,q_t(x,t)$$

where $\alpha(x)$ is a unit conversion factor from $\mathrm{mm} \cdot \Delta t^{-1}$ to $\mathrm{m^3 \cdot s^{-1}}$ for a single cell.

with $Q$ the surface discharge, $q_t$ the elemental discharge and $\Omega_x$ a 2D spatial domain that corresponds to all upstream cells flowing into cell $x$, i.e. the whole upstream catchment. Note that $\Omega_x$ is a subset of $\Omega$, $\Omega_x \subset \Omega$ and for the most upstream cells, $\Omega_x = \emptyset$.

     ○ *lr*

This routing operator is using a linear reservoir to rout upstream discharge to downstream following the drainage plan.

**Upstream Discharge**

Same as *lag0* Upstream Discharge

**Surface Discharge**

Update the routing reservoir state $h_{lr}$:

$$h_{lr}(x,t^*) = h_{lr}(x,t) + \frac{1}{\beta(x)}q_{up}(x,t)$$

where $\beta(x)$ is a conversion factor from $\mathrm{mm} \cdot \Delta t^{-1}$ to $\mathrm{m^3 \cdot s^{-1}}$ for the entire upstream domain $\Omega_x$.

Compute the routed discharge $q_{rt}$:

$$q_{rt}(x,t) = h_{lr}(x,t^*)\left(1 - \exp\left(\frac{-\Delta t}{60 \times l_{lr}}\right)\right)$$

Update the routing reservoir state $h_{lr}$:

$$h_{lr}(x,t) = h_{lr}(x,t^*) - q_{rt}(x,t)$$

Compute the surface discharge $Q$:

$$Q(x,t) = \beta(x)q_{rt}(x,t) + \alpha(x)q_t(x,t)$$

where $\alpha(x)$ is a conversion factor from $\mathrm{mm} \cdot \Delta t^{-1}$ to $\mathrm{m^3 \cdot s^{-1}}$ for a single cell.

     ○ *kw*

This routing operator is based on a conceptual 1D kinematic wave model that is numerically solved with a linearized implicit

numerical scheme (Chow et al., 1998). This is applicable given the drainage plan $\mathcal{D}_\Omega(x)$ that enables reducing the routing
problem to 1D.

The kinematic wave model is a simplification of the one-dimensional Saint-Venant hydraulic equations.

First, the mass conservation equation is written as:

$$\frac{\partial A}{\partial t} + \frac{\partial Q}{\partial x} = q \tag{D1}$$

where $\partial_\square$ denotes partial differentiation with respect to time or space, $A$ is the cross-sectional flow area, $Q$ is the discharge,
and $q$ represents lateral inflows.

The momentum equation is simplified by assuming that the water surface slope equals the bed slope, i.e., the flow is locally
uniform and gradually varied:

$$S_0 = S_f \tag{D2}$$

where $S_0$ is the bed slope and $S_f$ is the friction slope. This implies that the energy grade line is parallel to the channel bottom.
This simplification leads to an empirical relation between discharge and flow area or depth, as described by Chow et al. (1998).

$$A = a_{\text{kw}} Q^{b_{\text{kw}}} \tag{D3}$$

where $a_{\text{kw}}$ and $b_{\text{kw}}$ are two empirical constants that can also be related to the Manning friction law.

Injecting the parameterization from Eq. (D3) into the mass conservation equation (Eq. (D1)) yields the following one-equation
form of the kinematic wave model (Chow et al., 1998):

$$\frac{\partial Q}{\partial x} + a_{\text{kw}} b_{\text{kw}} Q^{b_{\text{kw}}-1} \frac{\partial Q}{\partial t} = q \tag{D4}$$

For the sake of clarity, the following variables are renamed for this section and the finite difference numerical scheme:

**Table D1.** Renamed variables

| Before | After |
|---|---|
| $Q(x,t)$ | $Q_i^j$ |
| $Q(x,t-1)$ | $Q_i^{j-1}$ |
| $q_t(x,t)$ | $q_i^j$ |
| $q_t(x,t-1)$ | $q_i^{j-1}$ |

**Upstream Discharge**

Same as *lag0* Upstream Discharge with $q_{up}$ denoted $Q_{i-1}^{j}$.

**Surface discharge**

Compute the intermediate variables $d_1$ and $d_2$

$$d_1 = \frac{\Delta t}{\Delta x}$$

$$d_2 = a_{kw} b_{kw} \left( \frac{Q_i^{j-1} + Q_{i-1}^{j}}{2} \right)^{b_{kw}-1}$$

Compute the intermediate variables $n_1$, $n_2$ and $n_3$

$$n_1 = d_1 Q_{i-1}^{j}$$

$$n_2 = d_2 Q_i^{j-1}$$

$$n_3 = d_1 \frac{q_i^{j-1} + q_i^{j}}{2}$$

Compute the surface discharge $Q_i^{j}$

$$Q_i^{j} = Q(x,t) = \frac{n_1 + n_2 + n_3}{d_1 + d_2}$$

with $Q$ the surface discharge, $q_t$ the elemental discharge, $a_{kw}$ the alpha kinematic wave parameter, $b_{kw}$ the beta kinematic wave parameter and $\Omega_x$ a 2D spatial domain that corresponds to all upstream cells flowing into cell $x$. Note that $\Omega_x$ is a subset of $\Omega$, $\Omega_x \subset \Omega$ and for the most upstream cells, $\Omega_x = \emptyset$.

**Appendix E: CPU Information**

```
Architecture:            x86_64
  CPU op-mode(s):        32-bit, 64-bit
  Address sizes:         48 bits physical, 48 bits virtual
  Byte Order:            Little Endian
CPU(s):                  192
  On-line CPU(s) list:   0-191
Vendor ID:               AuthenticAMD
  Model name:            AMD EPYC 7643 48-Core Processor
    CPU family:          25
    Model:               1
```

```
    Thread(s) per core:   2
    Core(s) per socket:   48
    Socket(s):            2
    Stepping:             1
Frequency boost:      enabled
    CPU max MHz:          2300.0000
    CPU min MHz:          1500.0000
    BogoMIPS:             4591.48
Virtualization features:
Virtualization:       AMD-V
Caches (sum of all):
  L1d:                    3 MiB (96 instances)
  L1i:                    3 MiB (96 instances)
  L2:                     48 MiB (96 instances)
L3:                   512 MiB (16 instances)
NUMA:
  NUMA node(s):           2
  NUMA node0 CPU(s):      0-47,96-143
  NUMA node1 CPU(s):      48-95,144-191
```

*Author contributions.* FC: lead developer of `smash` v1.0, conceptualization, numerical experiments and results analysis, manuscript preparation. NNTH: main developer of `smash` v1.0, conceptualization, results analysis, manuscript preparation. PAG: co-developer, conceptualization, research plan and supervision, results analysis, manuscript preparation, funding. MJA: co-developer of `smash` v1.0, main developer of the first wrapping and differentiable code, manuscript review. DO: main developer of the first Fortran code, manuscript review. BR: co-developer of `smash` v1.0, results analysis, research co-supervision, manuscript review. TDF, AEB, JD: contribution to co-developpement of 815 `smash` v1.0, manuscript review. PJ: results analysis, manuscript review, funding.

*Competing interests.* The authors declare no competing interests

*Acknowledgements.* The French national flood forecasting center, Service Central Vigicrues (ex. SCHAPI), is greatly acknowledged for funding research, software development and operational application, for long term collaboration on flood forecasting and data sharing. This work was also supported by funding from ANR grant ANR-21-CE04-0021-01 (MUFFINS project, "MUltiscale Flood Forecasting with

INnovating Solutions"). During the preparation of this work, the authors used Mistral AI in order to correct and improve English language. After using this tool, the authors reviewed and edited the content as needed and takes full responsibility for the content of the publication.

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
