# Peer review of "SMASH v1.0: A Differentiable and Regionalizable High-Resolution Hydrological Modeling and Data Assimilation Framework"

_EGUsphere, 2025_

## Referee Comment (RC1)

**General Comments**

Colleoni el al. develop a differentiable hydrological modeling framework, smash v1.0. They demonstrated good performance of the smash after conducting spatially distributed calibration and regionalization by learning the relationship between physical descriptors and parameters, with median evaluation metric KGE > 0.6. They tested their model framework in two configurations: (1) CONUS at 3km spatial resolution and (2) France watershed at 500m spatial resolution. Both applications show significant improvement in the model performance after using the regionalized parameter values. This study represents a significant contribution to hydrological modeling as it addressed a critical gap in hydrological model calibration, such as the traditional calibration method requiring tens of thousands of simulations with perturbed parameter values, which are computationally expensive. Please find my comments in the following. I recommend a revision before publication.

**Major Comments**

The description of the hydrological model is not clear. Could the authors provide a brief description of what processes are included in the hydrological model? If there is no space in the main text, the authors should consider adding it to the supplementary text.

What is the assumption of the routing model in smash? There exist different types of routing models. For example, large-scale 1D river routing models are commonly applied at relatively coarse spatial resolution with the assumption that each grid cell contains a representative channel. A 2-dimensional routing model that solves kinematic, diffusion, or dynamic wave routing at high spatial resolution, such as less than 1km, does not need such an assumption. However, it may not be appropriate to apply such 2-dimensional routing model at coarse spatial resolution. Which type of routing model was implanted in smash? It has been applied at 3km resolution for CONUS and 500m resolution for France application. It would be helpful if the authors could elaborate on the routing method.

Could the author report the computation time for the regionalization simulations vs. an individual forward run? This can help the readers to better understand the benefits of differentiable models. For example, in typical hydrological model calibrations, one has to run the model many times, e.g., 1000, with perturbed parameters, to obtain the optimal parameters. I think the differentiable model requires much fewer simulations to get the optimal parameter values, making its application to large scales more feasible.

There is no description of the calibrated parameters and their possible range. In addition, how are the parameters perturbed to solve the adjoint of the forward model in Figure 2 is

not clear. Are the parameters perturbed at watershed level or are they perturbed at grid level within each watershed?

**Specific Comments**

Line 61: Please provide the full name of MPR.

Line 60 – Line 65: There are more large sample studies in terms of calibration of spatially distributed models using efficient statistical approaches. For example,

Yang, Y., Pan, M., Beck, H. E., Fisher, C. K., Beighley, R. E., Kao, S. C., Hong, Y., and Wood, E. F.: In quest of calibration density and consistency in hydrologic modeling: distributed parameter calibration against streamflow characteristics, Water. Resour Res., 55, 7784–7803, 2019.
Xu, D., Bisht, G., Sargsyan, K., Liao, C., and Leung, L. R.: Using a surrogate-assisted Bayesian framework to calibrate the runoff-generation scheme in the Energy Exascale Earth System Model (E3SM) v1, Geosci. Model Dev., 15, 5021–5043, https://doi.org/10.5194/gmd-15-5021-2022, 2022.

Hirpa, F. A., Salamon, P., Beck, H. E., Lorini, V., Alfieri, L., Zsoter, E., and Dadson, S. J.: Calibration of the Global Flood Awareness System (GloFAS) using daily streamflow data, J. Hydrol., 566, 595–606, https://doi.org/10.1016/j.jhydrol.2018.09.052, 2018.

Line 98: What type of mesh does smash support? Structured or unstructured?

Line 114 – Line 129: Are the snow, hydrological, and routing operators process-based functions or data-driven operators?

Line 159: Is equal weight used for w_g? Please specify how w_g was estimated for each gauge. In addition, how j_reg was estimated?

Line 215: Which year was chosen?

Line 221: Please specify how the computation times were evaluated. According to my experience, it is unrealistic for a model to take only 0.1 seconds to run 365 time steps for 3846 grid cells. This relates to my previous comments: are the snow, hydrological, and routing operators process-based models?

Line 226: The better scaling in the inverse run is likely due to it being more computationally expensive than the forward run since the inverse run requires more computations.

Line 229: Why the routing scheme not be parallelized? There are a lot of global river routing models, and 2-dimensional hydrodynamic models are well parallelized.

Figure 5: Please add longitude and latitude to the figure. Also, consider adding a colorbar to show what is plotted.

Line 268: Which 10 years are used for "warm-up"?

Line 270: What parameters are calibrated?

Line 275 – Line 280: Can you also report the default forward model performance?

Line 279: Please show the calibrated parameter values. Also, for the spatially uniform parameters, are they spatially uniform for the whole CONUS? Or are they only spatially uniform for each basin?

Line 279: I found the performance between spatially uniform and spatially distributed simulations is very close.

Line 341: For the non-inertial shallow water models, do you mean diffusion wave routing?

Line 342- 344: I wonder if the authors can give some comments on how to implement the differentiable model to more complex land surface models or hydrological models?

Line 368: Why change the evaluation metric from KGE to NSE in the application of Frace?

Line 403: I agree with the authors that the smash framework represents an advancement in data assimilation in the hydrological model. However, I disagree that the smash framework represents an advancement in the hydrological model. Based on the method section, smash implements simplified hydrological processes.

---

## Author Response (AR1)

**Authors' Response to Editors/Reviewers of**

**SMASH v1.0: A Differentiable and Regionalizable High-Resolution Hydrological Modeling and Data Assimilation Framework**

*GMD,*
* * *
**RC:** *Reviewers' Comment*,     AR: Authors' Response,     ☐ Manuscript Text

Dear Editors and Reviewers,

We greatly appreciate your time and effort in handling our manuscript. We extend special thanks to the reviewers for their thorough and constructive comments, which have significantly improved our work. Below, we provide a point-by-point response addressing the reviewers' concerns.

We hope these revisions have strengthened our manuscript and made it more suitable for publication in GMD.

François Colleoni and Pierre-André Garambois, on behalf of the authors.

**1. Reviewer 1**

**1.1. Major comments**

**RC:** *The description of the hydrological model is not clear. Could the authors provide a brief description of what processes are included in the hydrological model? If there is no space in the main text, the authors should consider adding it to the supplementary text.*

AR: Thank you for the suggestion. We will clarify the description of the different model structures in the revised manuscript and strengthen the reference to the detailed online documentation available at https://smash.recover.inrae.fr. While it would be possible to replicate the full model structure in the supplementary material, we believe this is unnecessary given that the online documentation provides comprehensive and openly accessible information on the hydrological operators, which are classical but employed in an original way in our framework.

> The operators available in smash are listed above, and further detailed in Appendix D and in the online documentation (https://smash.recover.inrae.fr/math_num_documentation/forward_structure.html).
>
> ○ *zero*
> This snow operator simply means that there is no snow operator.
>
> $$m_{lt}(x, t) = 0$$
>
> with $m_{lt}$ the melt flux.
>
> ...

**RC:** *What is the assumption of the routing model in smash? There exist different types of routing models. For example, large-scale 1D river routing models are commonly applied at relatively coarse spatial resolution with the assumption that each grid cell contains a representative channel. A 2-dimensional routing model that solves kinematic, diffusion, or dynamic wave routing at high spatial resolution, such as less than 1km, does not need such an assumption. However, it may not be appropriate to apply such 2-dimensional routing*

*model at coarse spatial resolution. Which type of routing model was implanted in smash? It has been applied at 3km resolution for CONUS and 500m resolution for France application. It would be helpful if the authors could elaborate on the routing method.*

AR:   We will clarify in the revised manuscript that the routing module implemented in smash is a classical 1D conceptual kinematic wave model, applied over a D8 flow direction network. This reduces the routing problem to a 1D structure along predefined flow paths. The model is solved numerically using a linearized implicit scheme (Chow et al., 1998). Despite its simplicity, this approach has shown good performance and robustness in large-sample applications, both at 3 km resolution over CONUS and 500 m resolution over France.

> A classical 1D conceptual kinematic wave model, applied over a D8 drainage plan $\mathcal{D}_\Omega$ without channel. The model is solved numerically using a linearized implicit scheme (Chow et al., 1998).

RC:   *Could the author report the computation time for the regionalization simulations vs. an individual forward run? This can help the readers to better understand the benefits of differentiable models. For example, in typical hydrological model calibrations, one has to run the model many times, e.g., 1000, with perturbed parameters, to obtain the optimal parameters. I think the differentiable model requires much fewer simulations to get the optimal parameter values, making its application to large scales more feasible.*

AR:   We will report the computation times for the regionalization simulation, particularly for the most computationally demanding configuration using the ANN, as well as the corresponding forward run times. For reference, the full regionalization procedures (including calibration, input, and output operations) already take approximately 95 hours for the CONUS case and 31 hours for the Aude case. We will additionally run and report the computation times of the associated forward simulations.

> Regarding computation times, the calibration with ANN mapping over periods $p1$ and $p2$ took 95 hours. This calibration involved 350 iterations, corresponding to 350 calls to the adjoint model, and was performed using 16 threads. For comparison, a single adjoint model run takes approximately 16 minutes, whereas a direct model run takes around 5 minutes using the same number of threads.

> Regarding computation times, the calibration with ANN mapping over period p1 took 31 hours. This calibration involved 350 iterations, corresponding to 350 calls to the adjoint model, and was performed using 10 threads. For comparison, a single adjoint model run takes approximately 6 minutes, whereas a direct model run takes around 1 minute and 30 seconds using the same number of threads

RC:   *There is no description of the calibrated parameters and their possible range. In addition, how are the parameters perturbed to solve the adjoint of the forward model in Figure 2 is not clear. Are the parameters perturbed at watershed level or are they perturbed at grid level within each watershed?*

AR:   Thank you for pointing this out. We will include the calibration parameter ranges for all tested model structures in an appendix, in the form of a table listing parameter names and their respective ranges. A link to the online documentation will also be provided to clarify the meaning and role of each parameter, so as to avoid overloading the main text.

> A description of the calibrated parameters is provided in the Appendix {A, B, C}

Regarding the computation of cost function derivatives with respect to the parameters, these are not obtained through perturbation methods. Instead, they are computed accurately using the numerical adjoint model, derived via automatic code differentiation as described in the manuscript. This approach is more suitable for high-dimensional and nonlinear models, where perturbation methods tend to be less reliable. We will make this explanation more explicit in the revised version.

> smash features an efficient, differentiable Fortran solver using Tapenade to automatically derive the adjoint model that supports CPU forward-inverse parallel computing and spatially distributed optimization of large parameter vectors thanks to accurate cost gradient, interfaced in Python using f90wrap
>
> Flowchart of the inverse algorithm that uses $\nabla_\rho J$ the cost gradient with respect to the tunable control parameter $\rho$ obtained by solving the adjoint model $D_\rho \mathcal{M}$ of the forward model $\mathcal{M}$ obtained by automatic source code differentiation and enabling accurate gradient computation

Finally, the calibration can be performed in various modes: (i) uniform calibration with a single parameter set per watershed, (ii) fully distributed calibration with one parameter set per grid cell, or (iii) constrained calibration using physical descriptor maps for example, using binary masks over sub-basins to enable semi-distributed parameter configurations.

**1.2. Specific comments**

**RC:** *Line 61: Please provide the full name of MPR.*

AR: We will provide the full name of MPR.

> Large sample studies have also been undertaken with spatially distributed models among which VIC (Mizukami et al., 2017) with a Multiscale Parameter Regionalization (MPR) (Samaniego et al., 2010)

**RC:** *Line 60 – Line 65: There are more large sample studies in terms of calibration of spatially distributed models using efficient statistical approaches. For example,*

*Yang, Y., Pan, M., Beck, H. E., Fisher, C. K., Beighley, R. E., Kao, S. C., Hong, Y., and Wood, E. F.: In quest of calibration density and consistency in hydrologic modeling: distributed parameter calibration against streamflow characteristics, Water. Resour Res., 55, 7784–7803, 2019.*

*Xu, D., Bisht, G., Sargsyan, K., Liao, C., and Leung, L. R.: Using a surrogate-assisted Bayesian framework to calibrate the runoff-generation scheme in the Energy Exascale Earth System Model (E3SM) v1, Geosci. Model Dev., 15, 5021–5043, https://doi.org/10.5194/gmd-15-5021-2022, 2022.*

*Hirpa, F. A., Salamon, P., Beck, H. E., Lorini, V., Alfieri, L., Zsoter, E., and Dadson, S. J.: Calibration of the Global Flood Awareness System (GloFAS) using daily streamflow data, J. Hydrol., 566, 595–606, https://doi.org/10.1016/j.jhydrol.2018.09.052, 2018.*

AR: We will expand the bibliography in this section.

> Large sample studies have also been undertaken with spatially distributed models among which VIC (Mizukami et al., 2017) with a Multiscale Parameter Regionalization (MPR) (Samaniego et al., 2010) or with pixel-wise calibration on global maps of streamflow characteristics (Yang et al., 2019), a gridded version of HBV applied with MPR like descriptors-to-parameters regressions on a global dataset (Beck et al., 2020), GloFas (Hirpa et al., 2018), NHM (Towler et al., 2023), Wflow (Aerts et al., 2022; van Verseveld et al., 2024) or runoff relevant parameters of E3SM using a surrogate-assisted Bayesian framework (Xu et al., 2022).

**RC:** *Line 98: What type of mesh does smash support? Structured or unstructured?*

AR: smash only supports structured mesh.

> It is designed to simulate discharge hydrographs and hydrological states at any spatial location within a structured mesh and it ...

**RC:** *Line 114 – Line 129: Are the snow, hydrological, and routing operators process-based functions or data-driven operators?*

AR: All operators in smash version 1.0 snow, hydrological, and routing are classical process-based conceptual models. We will clarify this point in the revised manuscript and include a reminder of the link to our detailed online documentation for further reference.

> Several process-based conceptual operators are available in smash for composing a model

**RC:** *Line 159: Is equal weight used for $w_g$? Please specify how $w_g$ was estimated for each gauge. In addition, how $j_{reg}$ was estimated?*

AR: Thank you for highlighting this point. We acknowledge that this part was not clearly described in the manuscript and will clarify it in the revised version. In this study, equal weights were assigned to each watershed, which corresponds to minimizing the average of the individual cost functions. Additionally, no regularization term ($j_{reg}$) was applied in this work.

> In this study, equal weights were assigned to each gauge (i.e. $w_g = 1/N_G$), which corresponds to minimizing the average of the individual cost functions. Additionally, no regularization term was applied.

**RC:** *Line 215: Which year was chosen?*

AR: The evaluation period selected for this study was from July 31, 2010 to July 31, 2011. This information will be added to the revised manuscript for clarity.

> We compare smash over 3 zones: Sardinia, Great Britain/Ireland and North America at a spatial resolution of 1'30" ($\sim$ 3km $\times$ 3km) over a period of 1 year, from July 31, 2010 to July 31, 2011, randomly chosen, at a daily time step.

**RC:** *Line 221: Please specify how the computation times were evaluated. According to my experience, it is unrealistic for a model to take only 0.1 seconds to run 365 time steps for 3846 grid cells. This relates to my previous comments: are the snow, hydrological, and routing operators process-based models?*

AR: The computation time reported in the manuscript corresponds to the average time for a pure forward or inverse run, excluding the time required for input and output processes. After re-evaluating the results, we confirm that this time reflects the computational cost of our relatively simple and efficient forward hydrological model, which involves only a small number of numerical operations per grid cell. Additionally, as mentioned earlier, the snow, hydrological, and routing operators are indeed process-based models.

**RC:** *Line 226: The better scaling in the inverse run is likely due to it being more computationally expensive than the forward run since the inverse run requires more computations.*

AR: We agree with your observation and this will be explicitly stated in the revised version.

> This difference highlights better thread scaling for the adjoint run, with a speedup of around 4 for a direct run and 7 for an adjoint run with 16 threads likely because the adjoint run is more computationally demanding than the forward run.

**RC:** *Line 229: Why the routing scheme not be parallelized? There are a lot of global river routing models, and 2-dimensional hydrodynamic models are well parallelized.*

**AR:** You are correct, and we apologize for the confusion. The routing scheme is indeed parallelized, and we will correct this in the manuscript. What we intended to convey is that the routing scheme cannot be fully parallelized over the entire spatial domain, as it must be solved in a sequential manner from upstream to downstream. This distinction will be clarified in the revised version.

> Since the time-stepping loop cannot be parallelized at all and the routing scheme cannot be fully parallelized at pixel scale over the entire spatial domain, as it must be solved in a sequential manner from upstream to downstream

**RC:** *Figure 5: Please add longitude and latitude to the figure. Also, consider adding a colorbar to show what is plotted.*

**AR:** We will add both the longitude and latitude axes to the figure, as well as a colorbar to clearly indicate that the variable plotted is flow accumulation.

> Figure 5

**RC:** *Line 268: Which 10 years are used for "warm-up"?*

**AR:** The "warm-up" period corresponds to the following:

- $p1$: from August 1, 1990, to July 31, 2000

- $p2$: from August 1, 1997, to July 31, 2007

> For each period, the 10 preceding years are used as model "warm-up"

**RC:** *Line 270: What parameters are calibrated?*

**AR:** A detailed description of the calibrated parameters and their respective optimization ranges has been added in the appendix for reference.

> A description of the calibrated parameters is provided in the Appendix {A, B, C}

**RC:** *Line 275 – Line 280: Can you also report the default forward model performance?*

**AR:** Reporting the performance of the default forward model is not particularly meaningful in the context of this conceptual hydrological model. Our "default" model refers to a spatially uniform calibration of the parameters, which serves as a baseline for comparison.

**RC:** *Line 279: Please show the calibrated parameter values. Also, for the spatially uniform parameters, are they spatially uniform for the whole CONUS? Or are they only spatially uniform for each basin?*

**AR:** Thank you for your comment. We will include statistics on the calibrated parameter values in the appendix. Regarding the spatially uniform parameters, they have been optimized independently for each watershed. As a result, we have a spatially uniform set of parameters within each watershed.

> Summary statistics of the calibrated parameters are provided in Appendix A
>
> 2 calibration mappings on each catchment are tested:
>
> ○ Uniform: spatially uniform parameters
>
> ○ Distributed: spatially distributed parameters

**RC:** *Line 279: I found the performance between spatially uniform and spatially distributed simulations is very close.*

**AR:** Indeed, for this daily and 3 km resolution model, the performance between spatially uniform and spatially distributed simulations is very close. However, we do observe a slight improvement in the $KGE$ when using a spatially distributed calibration compared to a uniform one. At higher spatio-temporal resolutions, the performance gap becomes more pronounced, with uniform parameters yielding lower performance than a distributed calibration. This was clearer in a similar study conducted with an hourly and kilometric model over France (cf. https://hal.science/hal-04989183/document).

**RC:** *Line 341: For the non-inertial shallow water models, do you mean diffusion wave routing?*

**AR:** The shallow water model we refer to includes the convective inertia term in the momentum equation, which allows for a simple finite difference solution.

**RC:** *Line 342- 344: I wonder if the authors can give some comments on how to implement the differentiable model to more complex land surface models or hydrological models?*

**AR:** Any hydrological or land surface model can be integrated into the smash framework, provided it is compatible with the constraints of automatic differentiation imposed by the Tapenade tool. In particular, the model must be written in a differentiable form and comply with Tapenade's supported syntax and structures. We provide technical guidance on how automatic differentiation is implemented within smash in the online documentation, in the section: Automatic Differentiation – Development Process Details (https://smash.recover.inrae.fr/contributor_guide/development_process_details.html#automatic-differentiation).

**RC:** *Line 368: Why change the evaluation metric from KGE to NSE in the application of France?*

**AR:** The change in evaluation metric from $KGE$ to $NSE$ in the France application serves two purposes: (i) to demonstrate that our advanced calibration and regionalization approaches can be used with different cost functions, and (ii) to focus on a case study with a relatively fine spatial and temporal resolution, where greater emphasis is placed on the high-water aspects. Hence, the use of $NSE$ as both a calibration and evaluation metric in this context.

> This section is similar to the previous one, using the same regionalization method, but with differences in the gauges selected for calibration and validation, as well as in the cost function. This section focuses on national data at a finer spatio-temporal scale, applying the method at the watershed level, which is more relevant for operational flood forecasting

**RC:** *Line 403: I agree with the authors that the smash framework represents an advancement in data assimilation in the hydrological model. However, I disagree that the smash framework represents an advancement in the hydrological model. Based on the method section, smash implements simplified hydrological processes.*

**AR:** We agree with your observation and will revise the sentence accordingly. We believe that the smash framework represents an advancement in modular, regionalizable, differentiable numerical modeling, as well as in hydrological data assimilation.

> The recently released smash framework represents an advancement in modular, regionalizable, differentiable numerical modeling, as well as in hydrological data assimilation.

**2. Reviewer 2**

**2.1. Minor comments**

**RC:** *Data assimilation is not implemented or validated in this study, so I suggest removing it from the title and abstract.*

**AR:**  The proposed framework adopts a variational data assimilation (VDA) approach, utilizing a numerically differentiable solver to enable accurate estimation of cost function gradients for solving the inverse problem. In the study, the inverse problem consists in searching parameter maps from discharge data, without or with descriptors-to-parameters regionalization mapping in the forward model (the so called Hybrid Data Assimilation and Parameter Regionalization (HDA-PR) approach Huynh et al. (2024)), over long time windows with the VDA algorithm. The smash framework also enables to perform VDA at shorter time scales and to infer initial states (not illustrated). These aspects will be clarified in the revised version, including a distinction between data assimilation for reanalysis and near real-time applications, and upgrades required for background and observation covariances matrices, aimed at inferring uncertain parameters—including initial states.

> The proposed differentiable and regionalizable spatially distributed modeling framework is designed for gradient-based variational data assimilation applicable to initial states (not shown) and parameters estimation at multiple time scales
>
> The proposed framework leverages adjoint-based variational data assimilation (VDA), enabling the simultaneous inference of high-dimensional and spatially distributed parameters (as illustrated) and initial states (implementation available and tested in smash v1.0 but not shown), applicable at both long and short time scales.

**RC:** *Introduction: Differentiable modeling for large sample and high-resolution hydrologic studies have seen notable progress in recent years. I encourage the authors to mention some related works to fully reflect recent progress in this area.*

**AR:**  We will expand the bibliography in this section.

> Differentiable numerical hydrological modeling has made significant progress in recent years (spatially distributed variational data assimilation in Castaings et al. (2009); Lee et al. (2012); Jay-Allemand et al. (2020)), for large catchment sample studies with hybrid physics-AI, both with lumped approaches (e.g., Feng et al. (2024)) and with high-resolution, spatially distributed frameworks (Huynh et al., 2024, 2025).

**RC:** *Line 118-120: What does GR stand for? It needs to be explained above.*

**AR:**  "GR" stands for "Génie Rural," which translates to "Rural Engineering" in English.

> *gr4*: Génie Rural (GR)-like module (Perrin et al., 2003; Mathevet, 2005)

**RC:** *Line Line 137: Typo "$[l_k, u_k]$"*

**AR:**  Thank you for pointing that out. We will correct the typo in the revised manuscript.

> a transformation based on a sigmoid function with values in $[l_k, u_k]$

**RC:** *Line 170: I think here should be the optimization/training problem rather than a typical variational data assimilation method, which typically uses near-real-time observations to update model states and forecasts. As mentioned above, data assimilation is not examined in this study.*

**AR:** VDA is traditionally used to infer uncertain or unknown model parameters and initial states from available data using gradient based optimization, which is enable by the proposed smash framework that incorporates a VDA algorithm. You are right, the primary focus of this study is on model parameters training in reanalysis, but using the VDA algorithm, rather than "real-time" data assimilation. This will be clarified.

**RC:** *Line 221-220: Is the model parallelized across grids using OpenMP? How many threads per CPU were used for parallel execution? Please clarify.*

**AR:** Thank you for your comment. Yes, the model is parallelized across the spatial grid using OpenMP. We made an error in referring to CPUs; in this case, only one thread is used per CPU. In the revised manuscript, we will consistently refer to threads rather than CPUs to avoid any confusion.

> *CPU replaced by thread*

**RC:** *Line 233-234: The inverse run includes an iteration loop from the optimization algorithm, and within each iteration, there is a time loop. Is this correct? Please clarify.*

**AR:** There is some confusion in the manuscript between the terms "inverse run" and "inverse routine." To clarify, an "inverse run" refers to a single call to the adjoint model during the optimization process. The number of adjoint model calls within each optimization iteration may vary depending on the optimization algorithm used, such as L-BFGS-B. On the other hand, the "inverse routine" described in Figure 2 refers to multiple calls to the adjoint model. To resolve this confusion, we will maintain the term "inverse routine" in Figure 2 and rename "inverse run" to "adjoint run" in the 'Computational performance' section.

> *inverse run replaced by adjoint run*

**RC:** *Figure 4: Instead of using the number of CPUS, it is more reasonable to use the number of cores/threads as x axis.*

**AR:** You are correct. We will make this adjustment in the revised manuscript.

> Figure 6

**RC:** *The difference between the experiments in Section 4.2 and Section 4.3 is not very clear. Please clarify.*

**AR:** The main distinction lies in the "precision" of the modeling approach. Both sections use the same regionalization method, with differences in the watersheds selected for calibration and validation, as well as the cost function. In Section 4.2, we use global data over a large spatio-temporal scale and apply the regionalization algorithm at a country-wide scale. In Section 4.3, we focus on national data at a finer spatio-temporal scale, applying the method at the watershed level, which is more relevant for operational flood forecasting.

> This section is similar to the previous one, using the same regionalization method, but with differences in the gauges selected for calibration and validation, as well as in the cost function. This section focuses on national data at a finer spatio-temporal scale, applying the method at the watershed level, which is more relevant for operational flood forecasting.

**References**

Aerts, J.P.M., Hut, R.W., van de Giesen, N.C., Drost, N., van Verseveld, W.J., Weerts, A.H., Hazenberg, P., 2022. Large-sample assessment of varying spatial resolution on the streamflow estimates of the wflow_sbm hydrological model. Hydrology and Earth System Sciences 26, 4407–4430. URL: https://hess.copernicus.org/articles/26/4407/2022/, doi:.

Beck, H.E., Pan, M., Lin, P., Seibert, J., van Dijk, A.I.J.M., Wood, E.F., 2020. Global fully distributed parameter regionalization based on observed streamflow from 4,229 headwater catchments. Journal of Geophysical Research: Atmospheres 125, e2019JD031485. URL: https://agupubs.onlinelibrary.wiley.com/doi/abs/10.1029/2019JD031485, doi:, arXiv:https://agupubs.onlinelibrary.wiley.com/doi/pdf/10.1029/2019JD031485. e2019JD031485 10.1029/2019JD031485.

Castaings, W., Dartus, D., Le Dimet, F.X., Saulnier, G.M., 2009. Sensitivity analysis and parameter estimation for distributed hydrological modeling: potential of variational methods. Hydrology and Earth System Sciences 13, 503 – 517.

Chow, V.T., Maidment, D.R., Mays, L.W., 1998. Applied Hydrology. McGraw-Hill Series in Water Resources and Environmental Engineering.

Feng, D., Beck, H., de Bruijn, J., Sahu, R.K., Satoh, Y., Wada, Y., Liu, J., Pan, M., Lawson, K., Shen, C., 2024. Deep dive into hydrologic simulations at global scale: harnessing the power of deep learning and physics-informed differentiable models ($\delta$hbv-globe1.0-hydrodl). Geoscientific Model Development 17, 7181–7198. URL: https://gmd.copernicus.org/articles/17/7181/2024/, doi:.

Hirpa, F.A., Salamon, P., Beck, H.E., Lorini, V., Alfieri, L., Zsoter, E., Dadson, S.J., 2018. Calibration of the global flood awareness system (glofas) using daily streamflow data. Journal of Hydrology 566, 595–606. URL: https://www.sciencedirect.com/science/article/pii/S0022169418307467, doi:.

Huynh, N.N.T., Garambois, P.A., Colleoni, F., Renard, B., Roux, H., Demargne, J., Jay-Allemand, M., Javelle, P., 2024. Learning regionalization using accurate spatial cost gradients within a differentiable high-resolution hydrological model: Application to the french mediterranean region. Water Resources Research 60, e2024WR037544. URL: https://agupubs.onlinelibrary.wiley.com/doi/abs/10.1029/2024WR037544, doi:, arXiv:https://agupubs.onlinelibrary.wiley.com/doi/pdf/10.1029/2024WR037544. e2024WR037544 2024WR037544.

Huynh, N.N.T., Garambois, P.A., Renard, B., Colleoni, F., Monnier, J., Roux, H., 2025. A distributed hybrid physics-ai framework for learning corrections of internal hydrological fluxes and enhancing high-resolution regionalized flood modeling. URL: https://egusphere.copernicus.org/preprints/2025/egusphere-2024-3665/, doi:.

Jay-Allemand, M., Javelle, P., Gejadze, I., Arnaud, P., Malaterre, P.O., Fine, J.A., Organde, D., 2020. On the potential of variational calibration for a fully distributed hydrological model: application on a mediterranean catchment. Hydrology and Earth System Sciences 24, 5519–5538.

Lee, H., Seo, D.J., Liu, Y., Koren, V., McKee, P., Corby, R., 2012. Variational assimilation of streamflow into operational distributed hydrologic models: effect of spatiotemporal scale of adjustment. Hydrology and Earth System Sciences 16, 2233–2251. URL: https://hess.copernicus.org/articles/16/2233/2012/, doi:.

Mathevet, T., 2005. Quels modeles pluie-debit globaux au pas de temps horaire? Développements empiriques et intercomparaison de modeles sur un large échantillon de bassins versants. Ph.D. thesis. Ph. D. thesis, ENGREF, 463 pp.

Mizukami, N., Clark, M.P., Newman, A.J., Wood, A.W., Gutmann, E.D., Nijssen, B., Rakovec, O., Samaniego, L., 2017. Towards seamless large-domain parameter estimation for hydrologic models. Water Resources Research 53, 8020–8040. URL: https://agupubs.onlinelibrary.wiley.com/doi/abs/10.1002/2017WR020401, doi:, arXiv:https://agupubs.onlinelibrary.wiley.com/doi/pdf/10.1002/2017WR020401.

Perrin, C., Michel, C., Andrèassian, V., 2003. Improvement of a parsimonious model for streamflow simulation. Journal of hydrology 279, 275–289.

Samaniego, L., Kumar, R., Attinger, S., 2010. Multiscale parameter regionalization of a grid-based hydrologic model at the mesoscale. Water Resources Research 46. URL: https://agupubs.onlinelibrary.wiley.com/doi/abs/10.1029/2008WR007327, doi:, arXiv:https://agupubs.onlinelibrary.wiley.com/doi/pdf/10.1029/2008WR007327.

Towler, E., Foks, S.S., Dugger, A.L., Dickinson, J.E., Essaid, H.I., Gochis, D., Viger, R.J., Zhang, Y., 2023. Benchmarking high-resolution hydrologic model performance of long-term retrospective streamflow simulations in the contiguous united states. Hydrology and Earth System Sciences 27, 1809–1825. URL: https://hess.copernicus.org/articles/27/1809/2023/, doi:.

van Verseveld, W.J., Weerts, A.H., Visser, M., Buitink, J., Imhoff, R.O., Boisgontier, H., Bouaziz, L., Eilander, D., Hegnauer, M., ten Velden, C., Russell, B., 2024. Wflow_sbm v0.7.3, a spatially distributed hydrological model: from global data to local applications. Geoscientific Model Development 17, 3199–3234. URL: https://gmd.copernicus.org/articles/17/3199/2024/, doi:.

Xu, D., Bisht, G., Sargsyan, K., Liao, C., Leung, L.R., 2022. Using a surrogate-assisted bayesian framework to calibrate the runoff-generation scheme in the energy exascale earth system model (e3sm) v1. Geoscientific Model Development 15, 5021–5043. URL: https://gmd.copernicus.org/articles/15/5021/2022/, doi:.

Yang, Y., Pan, M., Beck, H.E., Fisher, C.K., Beighley, R.E., Kao, S.C., Hong, Y., Wood, E.F., 2019. In quest of calibration density and consistency in hydrologic modeling: Distributed parameter calibration against streamflow characteristics. Water Resources Research 55, 7784–7803. URL: https://agupubs.onlinelibrary.wiley.com/doi/abs/10.1029/2018WR024178, doi:, arXiv:https://agupubs.onlinelibrary.wiley.com/doi/pdf/10.1029/2018WR024178.

---

## Referee Report (RR1)

**General comment:** I appreciate the authors' efforts in addressing my previous comments. I found most of my comments were well addressed. I recommend a minor revision with following additional comments for the authors to consider.

Regarding computation times, the calibration with ANN mapping over periods p1 and p2 took 95 hours. This calibration involved 350 iterations, corresponding to 350 calls to the adjoint model, and was performed using 16 threads. For comparison, a single adjoint model run takes approximately 16 minutes, whereas a direct model run takes around 5 minutes using the same number of threads.

Regarding computation times, the calibration with ANN mapping over period p1 took 31 hours. This calibration involved 350 iterations, corresponding to 350 calls to the adjoint model, and was performed using 10 threads. For comparison, a single adjoint model run takes approximately 6 minutes, whereas a direct model run takes around 1 minute and 30 seconds using the same number of threads.

**Comment:** Why 350 integrations is used for calibration?

You are correct, and we apologize for the confusion. The routing scheme is indeed parallelized, and we will correct this in the manuscript. What we intended to convey is that the routing scheme cannot be fully parallelized over the entire spatial domain, as it must be solved in a sequential manner from upstream to downstream. This distinction will be clarified in the revised version.

Since the time-stepping loop cannot be parallelized at all and the routing scheme cannot be fully parallelized at pixel scale over the entire spatial domain, as it must be solved in a sequential manner from upstream to downstream.

**Comment:** I cannot agree with the authors. Routing scheme can be fully parallelized over the entire spatial domain. A well paralleled river routing model does not need to be solved in a sequential manner from upstream to downstream.

Reporting the performance of the default forward model is not particularly meaningful in the context of this conceptual hydrological model. Our "default" model refers to a spatially uniform calibration of the parameters, which serves as a baseline for comparison.

**Comment:** I cannot agree with this response. It is critical to compare the calibrated model with the default model with default (not uniform calibration) parameter values. If there is no improvement from default parameters to calibrated parameters, then why spending efforts on make the model differentiable? I highly suggest the authors to report the performance of default model.

The shallow water model we refer to includes the convective inertia term in the momentum equation, which allows for a simple finite difference solution.

**Comment:** In this case, it is not non-inertial shallow water model. It is confusing.

Any hydrological or land surface model can be integrated into the smash framework, provided it is compatible with the constraints of automatic differentiation imposed by the Tapenade tool. In particular, the model must be written in a differentiable form and comply with Tapenade's supported syntax and structures. We provide technical guidance on how automatic differentiation is implemented within smash in the online documentation, in the section: Automatic Differentiation – Development Process Details.

**Comment:** My concern is that it is challenging to rewrite a complex hydrological model in a differential form. I am not aware of any example of differentiable model for complex hydrological models. Then the issue is why we need differentiable model for simplified hydrological model, which is computational cheap and can be easily run tens of thousands of times for calibration. It will be helpful for the authors to add some discussion on this limitation.

---

## Author Response (AR2)

**Authors' Response to Editors/Reviewers of**

**SMASH v1.0: A Differentiable and Regionalizable High-Resolution Hydrological Modeling and Data Assimilation Framework**

*GMD,*
* * *
**RC:** *Reviewers' Comment*,    AR: Authors' Response,    ☐ Manuscript Text

Dear Editors and Reviewers,

We greatly appreciate your time and effort in handling our manuscript. We extend special thanks to the reviewers for their thorough and constructive comments, which have significantly improved our work. Below, we provide a point-by-point response addressing the reviewers' concerns.

We hope these revisions have strengthened our manuscript and made it more suitable for publication in GMD.

François Colleoni and Pierre-André Garambois, on behalf of the authors.

**1. Reviewer 1**

**1.1. General comments**

**RC:** *I appreciate the authors' efforts in addressing my previous comments. I found most of my comments were well addressed. I recommend a minor revision with following additional comments for the authors to consider.*

AR:   Thanks

**1.2. Comment**

**RC:** *Why 350 integrations is used for calibration?*

AR:   The number of 350 iterations was somewhat arbitrary, chosen during the regionalization process using a neural network. The optimization algorithm employed does not rely on other stopping criteria besides the maximum number of iterations. In our case and based on our experiments and the behavior of the loss descent (Huynh et al., 2024b), this value was set to 350, which is sufficiently high to allow the optimizer to reach a reasonable level of convergence (Figure 1).

**RC:** *I cannot agree with the authors. Routing scheme can be fully parallelized over the entire spatial domain. A well paralleled river routing model does not need to be solved in a sequential manner from upstream to downstream.*

AR:   The sentence suggesting that full spatial parallelization is not possible will be revised to clarify that this limitation applies specifically to the routing model implemented in our case. We would be grateful if you could suggest us a more efficient routing approach.

> Although the time-stepping loop cannot be parallelized, and the routing scheme in `smash` must be solved sequentially from upstream to downstream, allowing only partial parallelization over the entire spatial domain, the approach still offers a substantial reduction in computation time.

**RC:** *I cannot agree with this response. It is critical to compare the calibrated model with the default model with*

[Figure]

Figure 1: The descent of the cost function for the CONUS case.

> ***default (not uniform calibration) parameter values. If there is no improvement from default parameters to calibrated parameters, then why spending efforts on make the model differentiable? I highly suggest the authors to report the performance of default model.***

AR: A conceptual model is only meaningful when properly calibrated. `smash` does not include physically valid default parameters, but instead uses numerical defaults only to ensure proper initialization. Model performance is systematically improved when calibrated parameters replace these initial values. To our knowledge, there are no hydrological modeling studies based on fully conceptual models that compare the performance of calibrated parameters versus "default" parameters. We do not believe that including such a comparison would add value to the article. However, a boxplot showing the performance with "default" parameters is provided in Figure 2. Moreover, although not explicitly mentioned in the paper, the spatially uniform calibration does not require the use of the adjoint model. Optimization is performed using a gradient-free optimizer (few details can be found here here). For this reason, we define the spatially uniform calibration as our performance baseline.

> ○ Uniform: spatially uniform parameters (gradient-free optimization)
>
> ○ Distributed: spatially distributed parameters (gradient-based optimization)
>
> ○ Multi-Linear: a multiple linear regression is used as transfer function from descriptors to spatialized parameters (gradient-based optimization)
>
> ○ ANN: a multi-layer perceptron composed of 3 hidden layers is used as a transfer function from descriptors to spatialized parameters (gradient-based optimization)

**RC:** ***In this case, it is not non-inertial shallow water model. It is confusing.***

AR: You are right — there was a mistake in our previous response and in the manuscript. We plan to implement a dynamic wave model that neglects the convective acceleration term, but retains local acceleration and pressure gradient terms (Bates et al., 2010). We will clarify this point in the revised version to avoid confusion.

[Figure]

Figure 2: Comparison of the Kling-Gupta Efficiency ($KGE$) performance of different smash hydrological models with spatially uniform "default" parameters, spatially uniform and distributed calibrated parameters.

Further work focuses on enriching `smash` with hydraulic models, starting with 1D and 2D dynamic wave model that neglects the convective acceleration term, but retains local acceleration and pressure gradient terms (Bates et al., 2010) for numerical implementation simplicity.

**RC:** *My concern is that it is challenging to rewrite a complex hydrological model in a differential form. I am not aware of any example of differentiable model for complex hydrological models. Then the issue is why we need differentiable model for simplified hydrological model, which is computational cheap and can be easily run tens of thousands of times for calibration. It will be helpful for the authors to add some discussion on this limitation.*

AR: We agree that, for simple hydrological models with only a few parameters, making the model differentiable may not be essential. In such cases, parsimonious models with uniform spatial parameters, calibration in low-dimensional settings can be performed efficiently using optimization algorithms that do not require gradient information or through sampling-based approaches. However, when moving towards spatially distributed calibration or regionalization using neural networks (requiring gradients of spatially distributed model parameters (Huynh et al., 2024b), the situation changes significantly. These approaches typically involve high-dimensional optimization problems, with potentially several thousands of parameters (e.g., in the case studies presented in this article, the number of parameters reaches 8,774 and 4,613 for the CONUS and Aude cases, respectively). In such contexts, exhaustive sampling of the parameter space becomes computationally challenging, whereas adjoint-based methods are well suited for computing accurate cost function gradients with respect to large parameter vectors, enabling efficient gradient-based optimization. This approach is particularly effective for complex models with high-dimensional parameter spaces. We will include a discussion in the manuscript to clarify this point.

The differentiability of the forward numerical model is a key enabler for gradient-based optimization of high-dimensional parameter vectors. For example, in variational data assimilation for 1D or 2D hydraulic models (Monnier et al., 2016; Brisset et al., 2018), or in spatialized hydrology (Castaings et al., 2009; Jay-Allemand et al., 2020). While differentiability may appear unnecessary for simple lumped hydrological models with only a few parameters, where sampling-based calibration or gradient-free methods remain efficient, the situation changes drastically for spatially distributed models involving thousands of parameters. In such high-dimensional settings, exhaustive sampling becomes computationally infeasible. Numerical differentiability enables the computation of accurate gradients of the cost function or model outputs with respect to high-dimensional parameters, thereby facilitating the use of efficient gradient-based optimization methods. This is particularly important when coupling physical models with neural networks requiring accurate gradients, as demonstrated in recent work on learnable regionalization (Huynh et al., 2024b) and internal flux correction (Huynh et al., 2024a) within a spatially distributed model, with large-scale evaluations in (Huynh et al., 2025). These approaches rely on numerically differentiable solvers and accurate gradients enabling to train thousands of parameters effectively. This perspective aligns with Shen et al. (2023), who emphasizes the importance and potential of differentiable modeling in geosciences, highlighting how it can enhance learning, inference, and integration of physical knowledge within hybrid modeling frameworks.

**References**

Bates, P.D., Horritt, M.S., Fewtrell, T.J., 2010. A simple inertial formulation of the shallow water equations for efficient two-dimensional flood inundation modelling. Journal of Hydrology 387, 33–45. URL: https://www.sciencedirect.com/science/article/pii/S0022169410001538, doi:.

Brisset, P., Monnier, J., Garambois, P.A., Roux, H., 2018. On the assimilation of altimetric data in 1D Saint-Venant river flow models. Advances in water resources 119, 41–59. URL: https://doi.org/10.1016/j.advwatres.2018.06.004.

Castaings, W., Dartus, D., Le Dimet, F.X., Saulnier, G.M., 2009. Sensitivity analysis and parameter estimation for distributed hydrological modeling: potential of variational methods. Hydrology and Earth System Sciences 13, 503 – 517.

Huynh, N.N.T., Garambois, P.A., Colleoni, F., Renard, B., Monnier, J., Roux, H., 2024a. Multiscale learnable physical modeling and data assimilation framework: Application to high-resolution regionalized hydrological simulation of flash flood. Authorea Preprints doi:.

Huynh, N.N.T., Garambois, P.A., Colleoni, F., Renard, B., Roux, H., Demargne, J., Jay-Allemand, M., Javelle, P., 2024b. Learning regionalization using accurate spatial cost gradients within a differentiable high-resolution hydrological model: Application to the french mediterranean region. Water Resources Research 60, e2024WR037544. URL: https://agupubs.onlinelibrary.wiley.com/doi/abs/10.1029/2024WR037544, doi:, arXiv:https://agupubs.onlinelibrary.wiley.com/doi/pdf/10.1029/2024WR037544. e2024WR037544 2024WR037544.

Huynh, N.N.T., Garambois, P.A., Renard, B., Colleoni, F., Monnier, J., Roux, H., 2025. A distributed hybrid physics-ai framework for learning corrections of internal hydrological fluxes and enhancing high-resolution regionalized flood modeling. URL: https://egusphere.copernicus.org/preprints/2025/egusphere-2024-3665/, doi:.

Jay-Allemand, M., Javelle, P., Gejadze, I., Arnaud, P., Malaterre, P.O., Fine, J.A., Organde, D., 2020. On the potential of variational calibration for a fully distributed hydrological model: application on a mediterranean catchment. Hydrology and Earth System Sciences 24, 5519–5538.

Monnier, J., Couderc, F., Dartus, D., Larnier, K., Madec, R., Vila, J.P., 2016. Inverse algorithms for 2D shallow water equations in presence of wet dry fronts: Application to flood plain dynamics. Advances in Water Resources 97, 11–24. URL: https://doi.org/10.1016/j.advwatres.2016.07.005.

Shen, C., Appling, A.P., Gentine, P., Bandai, T., Gupta, H., Tartakovsky, A., Baity-Jesi, M., Fenicia, F., Kifer, D., Li, L., Liu, X., Ren, W., Zheng, Y., Harman, C.J., Clark, M., Farthing, M., Feng, D., Kumar, P., Aboelyazeed, D., Rahmani, F., Song, Y., Beck, H.E., Bindas, T., Dwivedi, D., Fang, K., Höge, M., Rackauckas, C., Mohanty, B., Roy, T., Xu, C., Lawson, K., 2023. Differentiable modelling to unify machine learning and physical models for geosciences. Nature Reviews Earth & Environment 4, 552–567. URL: https://doi.org/10.1038/s43017-023-00450-9, doi:.